# Distributionally Robust Bayesian Optimization with $\varphi$-divergences

**Hisham Husain**
Amazon
hushisha@amazon.com

**Vu Nguyen**
Amazon
vutngn@amazon.com

**Anton van den Hengel**
Amazon
hengelah@amazon.com

## Abstract

The study of robustness has received much attention due to its inevitability in data-driven settings where many systems face uncertainty. One such example of concern is Bayesian Optimization (BO), where uncertainty is multi-faceted, yet there only exists a limited number of works dedicated to this direction. In particular, there is the work of Kirschner et al. [26], which bridges the existing literature of Distributionally Robust Optimization (DRO) by casting the BO problem from the lens of DRO. While this work is pioneering, it admittedly suffers from various practical shortcomings such as finite contexts assumptions, leaving behind the main question *Can one devise a computationally tractable algorithm for solving this DRO-BO problem*? In this work, we tackle this question to a large degree of generality by considering robustness against data-shift in $\varphi$-divergences, which subsumes many popular choices, such as the $\chi^2$-divergence, Total Variation, and the extant Kullback-Leibler (KL) divergence. We show that the DRO-BO problem in this setting is equivalent to a finite-dimensional optimization problem which, even in the continuous context setting, can be easily implemented with provable sublinear regret bounds. We then show experimentally that our method surpasses existing methods, attesting to the theoretical results.

## 1 Introduction

Bayesian Optimization (BO) [29, 25, 52, 49, 34] allows us to model a black-box function that is expensive to evaluate, in the case where noisy observations are available. Many important applications of BO correspond to situations where the objective function depends on an additional context parameter [27, 57], for example in health-care, recommender systems can be used to model information about a certain type of medical domain. BO has naturally found success in a number of scientific domains [56, 20, 30, 18, 55] and also a staple in machine learning for the crucial problem of hyperparameter tuning [44, 36, 40, 41, 59].

As with all data-driven approaches, BO is prone to cases where the given data *shifts* from the data of interest. While BO models this in the form of Gaussian noise for the inputs to the objective function, the context distribution is assumed to be consistent. This can be problematic, for example in healthcare where patient information shifts over time. This problem exists in the larger domain of operations research under the banner of *distributionally robust optimization* (DRO) [46], where one is interested in being *robust* against shifts in the distribution observed. In particular, for a given *distance* between distributions D, DRO studies robustness against adversaries who are allowed to modify the observed distribution $p$ to another distribution in the set:

$$\{q : \mathsf{D}(p,q) \leq \varepsilon\},$$

for some $\varepsilon > 0$. One can interpret this as a ball of radius $\varepsilon$ for the given choice of D and the adversary perturbs the observed distribution $p$ to $q$ where $\varepsilon$ is a form of "budget".

37th Conference on Neural Information Processing Systems (NeurIPS 2023).

Distributional shift is a topical problem in machine learning and the results of DRO have been specialized in the context of supervised learning [12, 13, 11, 10, 7, 16, 22], reinforcement learning [21] and Bayesian learning [51], as examples. One of the main challenges however is that the DRO is typically intractable since in the general setting of continuous contexts, involves an infinite dimensional constrained optimization problem. The choice of D is crucial here as various choices such as the Wasserstein distance [6, 7, 9, 48], Maximum Mean Discrepancy (MMD) [53] and $\varphi$-divergences [1] [12, 13] allow for computationally tractable regimes. In particular, these specific choices of D have shown intimate links between regularization [22] which is a conceptually central topic of machine learning.

More recently however, DRO has been studied for the BO setting in Kirschner et al. [26], which as one would expect, leads to a complicated minimax problem, which causes a computational burden practically speaking. Kirschner et al. [26] makes the first step and casts the formal problem however develops an algorithm only in the case where D has been selected as the MMD. While, this work makes the first step and conceptualizes the problem of distributional shifts in context for BO, there are two main practical short-comings. Firstly, the algorithm is developed specifically to the MMD, which is easily computed, however cannot be replaced by another choice of D whose closed form is not readily accessible with samples such as the $\varphi$-divergence. Secondly, the algorithm is only tractable when the contexts are finite since at every iteration of BO, it requires solving an $M$-dimensional problem where $M$ is the number of contexts.

The main question that remains is, *can we devise an algorithm that is computationally tractable for tackling the DRO-BO setting*? We answer this question to a large degree of generality by considering distributional shifts against $\varphi$-divergences - a large family of divergences consisting of the extant Kullback-Leibler (KL) divergence, Total Variation (TV) and $\chi^2$-divergence, among others. In particular, we exploit existing advances made in the large literature of DRO to show that the BO objective in this setting for any choice of $\varphi$-divergence yields a computationally tractable algorithm, even for the case of continuous contexts. We also present a robust regret analysis that illustrates a sublinear regret. Finally, we show, along with computational tractability, that our method is empirically superior on standard datasets against several baselines including that of Kirschner et al. [26]. In summary, our main contributions are

1. A theoretical result showing that the minimax distributionally robust BO objective with respect to $\varphi$ divergences is equivalent to a single minimization problem.

2. An efficient algorithm, that works in the continuous context regime, for the specific cases of the $\chi^2$-divergence and TV distance, which admits a conceptually interesting relationship to regularization of BO.

3. A regret analysis that specifically informs how we can choose the DRO $\varepsilon$-budget to attain sublinear regret.

## 2 Related Work

Due to the multifaceted nature of our contribution, we discuss two streams of related literature, one relating to studies of robustness in Bayesian Optimization (BO) and one relating to advances in Distributionally Robust Optimization (DRO).

In terms of BO, the work most closest to ours is Kirschner et al. [26] which casts the distributionally robust optimization problem over contexts. In particular, the work shows how the DRO objective for any choice of divergence D can be cast, which is exactly what we build off. The main drawback of this method however is the limited practical setting due to the expensive inner optimization, which heavily relies on the MMD, and therefore cannot generalize easily to other divergences that are not available in closed forms. Our work in comparison, holds for a much more general class of divergences, and admits a practical algorithm that involves a finite dimensional optimization problem. In particular, we derive the result when D is chosen to be the $\chi^2$-divergence which we show performs the best empirically. This choice of divergence has been studied in the related problem of Bayesian quadrature [33], and similarly illustrated strong performance, complimenting our results. There also exists work of BO that aim to be robust by modelling adversaries through noise, point estimates or non-cooperative games [37, 32, 3, 39, 47]. The main difference between our work and theirs is

---

[1]as known as $f$-divergences in the literature

that the notion of robustness we tackle is at the *distributional* level. Another similar work to ours is that of Tay et al. [54] which considers approximating DRO-BO using Taylor expansions based on the sensitivity of the function. In some cases, the results coincide with ours however their result must account for an approximation error in general. Furthermore, an open problem as stated in their work is to solve the DRO-BO problem for continuous context domains, which is precisely one of the advantages of our work.

From the perspective of DRO, our work essentially is an extension of Duchi et al. [12, 13] which develops results that connect $\varphi$-divergence DRO to variance regularization. In particular, they assume $\varphi$ admits a continuous second derivative, which allows them to connect the $\varphi$-divergence to the $\chi^2$-divergence and consequently forms a general connection to constrained variance. While the work is pioneering, this assumption leaves out important $\varphi$-divergences such as the Total Variation (TV) - a choice of divergence which we illustrate performs well in comparison to standard baselines in BO. At the technical level, our derivations are similar to Ahmadi-Javid [2] however our result, to the best of our knowledge, is the first such work that develops it in the context of BO. In particular, our results for the Total Variation and $\chi^2$-divergence show that variance is a key penalty in ensuring robustness which is a well-known phenomena existing in the realm of machine learning [12, 11, 10, 22, 1].

## 3  Preliminaries

**Bayesian Optimization**  We consider optimizing a *black-box* function, $f : \mathcal{X} \to \mathbb{R}$ with respect to the *input* space $\mathcal{X} \subseteq \mathbb{R}^d$. As a black-box function, we do not have access to $f$ directly however receive input in a sequential manner: at time step $t$, the learner chooses some input $\mathbf{x}_t \in \mathcal{X}$ and observes the *reward* $y_t = f(\mathbf{x}_t) + \eta_t$ where the noise $\eta_t \sim \mathcal{N}(0, \sigma_f^2)$ and $\sigma_f^2$ is the output noise variance. Therefore, the goal is to optimize

$$\sup_{\mathbf{x} \in \mathcal{X}} f(\mathbf{x}).$$

Additional to the input space $\mathcal{X}$, we introduce the *context* spaces $\mathcal{C}$, which we assume to be compact. These spaces are assumed to be separable completely metrizable topological spaces.[2] We have a reward function, $f : \mathcal{X} \times \mathcal{C} \to \mathbb{R}$ which we are interested in optimizing with respect to $\mathcal{X}$. Similar to sequential optimization, at time step $t$ the learner chooses some input $\mathbf{x}_t \in \mathcal{X}$ and receives a context $c_t \in \mathcal{C}$ and $f(\mathbf{x}_t, c_t) + \eta_t$. Here, the learner can not choose a context $c_t$, but receive it from the environment. Given the context information, the objective function is written as

$$\sup_{\mathbf{x} \in \mathcal{X}} \mathbb{E}_{c \sim p}[f(\mathbf{x}, c)],$$

where $p$ is a probability distribution over contexts.

**Gaussian Processes**  We follow a popular choice in BO [49] to use GP as a surrogate model for optimizing $f$. A GP [43] defines a probability distribution over functions $f$ under the assumption that any subset of points $\{\mathbf{x}_i, f(\mathbf{x}_i)\}$ is normally distributed. Formally, this is denoted as:

$$f(\mathbf{x}) \sim \mathrm{GP}\left(m(\mathbf{x}), k(\mathbf{x}, \mathbf{x}')\right),$$

where $m(\mathbf{x})$ and $k(\mathbf{x}, \mathbf{x}')$ are the mean and covariance functions, given by $m(\mathbf{x}) = \mathbb{E}[f(\mathbf{x})]$ and $k(\mathbf{x}, \mathbf{x}') = \mathbb{E}\left[(f(\mathbf{x}) - m(\mathbf{x}))(f(\mathbf{x}') - m(\mathbf{x}'))^T\right]$. For predicting $f_* = f(\mathbf{x}_*)$ at a new data point $\mathbf{x}_*$, the conditional probability follows a univariate Gaussian distribution as $p\left(f_* \mid \mathbf{x}_*, [\mathbf{x}_1...\mathbf{x}_N], [y_1, ...y_N]\right) \sim \mathcal{N}\left(\mu(\mathbf{x}_*), \sigma^2(\mathbf{x}_*)\right)$. Its mean and variance are given by:

$$\mu(\mathbf{x}_*) = \mathbf{k}_{*,N} \mathbf{K}_{N,N}^{-1} \mathbf{y}, \qquad (1) \qquad \sigma^2(\mathbf{x}_*) = k_{**} - \mathbf{k}_{*,N} \mathbf{K}_{N,N}^{-1} \mathbf{k}_{*,N}^T \qquad (2)$$

where $k_{**} = k(\mathbf{x}_*, \mathbf{x}_*)$, $\mathbf{k}_{*,N} = [k(\mathbf{x}_*, \mathbf{x}_i)]_{\forall i \leq N}$ and $\boldsymbol{K}_{N,N} = [k(\mathbf{x}_i, \mathbf{x}_j)]_{\forall i, j \leq N}$. As GPs give full uncertainty information with any prediction, they provide a flexible nonparametric prior for Bayesian optimization. We refer to Rasmussen and Williams [43] for further details on GPs.

**Distributional Robustness**  Let $\Delta(\mathcal{C})$ denote the set of probability distributions over $\mathcal{C}$. A *divergence* between distributions $\mathrm{D} : \Delta(\mathcal{C}) \times \Delta(\mathcal{C}) \to \mathbb{R}$ is a dissimilarity measure that satisfies

---

[2]We remark that this is an extremely mild condition, satisfied by the large majority of considered examples.

$\Delta(p, q) \geq 0$ with equality if and only if $p = q$ for $p, q \in \Delta(\mathcal{C})$. For a function, $h : \mathcal{C} \to \mathbb{R}$, base probability measure $p \in \Delta(\mathcal{C})$, the central concern of Distributionally Robust Optimization (DRO) [4, 42, 5] is to compute

$$\sup_{q \in B_{\varepsilon, \mathsf{D}}(p)} \mathbb{E}_{q(c)}[h(c)], \tag{3}$$

where $B_{\varepsilon, \mathsf{D}}(p) = \{q \in \Delta(\mathcal{C}) : \mathsf{D}(p, q) \leq \varepsilon\}$, is ball of distributions $q$ that are $\varepsilon$ away from $p$ with respect to the divergence D. The objective in Eq. (3) is intractable, especially in setting where $\mathcal{C}$ is continuous as it amounts to a constrained infinite dimensional optimization problem. It is also clear that the choice of D is crucial for both computational and conceptual purposes. The vast majority of choices typically include the Wasserstein due to the transportation-theoretic interpretation and with a large portion of existing literature finding connections to Lipschitz regularization [6, 7, 9, 48]. Other choices where they have been studied in the supervised learning setting include the Maximum Mean Discrepancy (MMD) [53] and $\varphi$-divergences [12, 13].

**Distributionally Robust Bayesian Optimization**   Recently, the notion of DRO has been applied to BO [26, 54], who consider robustness with respect to shifts in the context space and therefore are interested in solving

$$\sup_{\mathbf{x} \in \mathcal{X}} \inf_{q \in B_{\varepsilon, D}(p)} \mathbb{E}_{c \sim q}[f(\mathbf{x}, c)],$$

where $p$ is the reference distribution. This objective becomes significantly more difficult to deal with since not only does it involve a constrained and possibly infinite dimensional optimization problem however also involves a minimax which can cause instability issues if solved iteratively.

Kirschner et al. [26] tackle these problems by letting D be the kernel Maximum Mean Discrepancy (MMD), which is a popular choice of discrepancy motivated by kernel mean embeddings [19]. In particular, the MMD can be efficiently estimated in $O(n^2)$ where $n$ is the number of samples. Naturally, this has two main drawbacks: The first is that it is still computationally expensive since one is required to solve two optimization problems, which can lead to instability and secondly, the resulting algorithm is limited to the scheme where the number of contexts is finite. In our work, we consider D to be a $\varphi$-divergence, which includes the Total Variance, $\chi^2$ and Kullback-Leibler (KL) divergence and furthermore show that minmax objective can be reduced to a single maximum optimization problem which resolves both the instability and finiteness assumption. In particular, we also present a similar analysis, showing that the robust regret decays sublinearly for the right choices of radii.

## 4   $\varphi$-Robust Bayesian Optimization

In this section, we present the main result on distributionally robustness when applied to BO using $\varphi$-divergence. Therefore, we begin by defining this key quantity.

**Definition 1 ($\varphi$-divergence)** *Let $\varphi : \mathbb{R} \to (-\infty, \infty]$ be a convex, lower semi-continuous function such that $\varphi(1) = 0$. The $\varphi$-divergence between $p, q \in \Delta(\mathcal{C})$ is defined as*

$$\mathsf{D}_\varphi(p, q) = \mathbb{E}_{q(c)} \left[ \varphi \left( \frac{dp}{dq}(c) \right) \right],$$

*where $dp/dq$ is the Radon-Nikodym derivative if $p \ll q$ and $\mathsf{D}_\varphi(p, q) = +\infty$ otherwise.*

Popular choices of the convex function $\varphi$ include $\varphi(u) = (u - 1)^2$ which yields the $\chi^2$ and, $\varphi(u) = |u - 1|$, $\varphi(u) = u \log u$ which correspond to the $\chi^2$ and KL divergences respectively. At any time step $t \geq 1$, we consider distributional shifts with respect to an $\varphi$-divergence for any choice of $\varphi$ and therefore relevantly define the DRO ball as

$$B_\varphi^t(p_t) := \{q \in \Delta(\mathcal{C}) : \mathsf{D}_\varphi(q, p_t) \leq \varepsilon_t\},$$

where $p_t = \frac{1}{t} \sum_{s=1}^t \delta_{c_s}$ is the reference distribution and $\varepsilon_t$ is the distributionally robust radius chosen at time $t$. We remark that for our results, the choice of $p_t$ is flexible and can be chosen based on the specific domain application. The $\varphi$ divergence, as noted from the definition above, is only defined

finitely when the measures $p, q$ are absolutely continuous to each other and there is regarded as a *strong* divergence in comparison to the Maximum Mean Discrepancy (MMD), which is utilized in Kirschner et al. [26]. The main consequence of this property is that the geometry of the ball $B_\varphi^t$ would differ based on the choice of $\varphi$-divergence. The $\varphi$-divergence is a very popular choice for defining this ball in previous studies of DRO in the context of supervised learning due to the connections and links it has found to variance regularization [12, 13, 11].

We will exploit various properties of the $\varphi$-divergence to derive a result that reaps the benefits of this choice such as a reduced optimization problem - a development that does not currently exist for the MMD [26]. We first define the convex conjugate of $\varphi$ as $\varphi^\star(u) = \sup_{u' \in \text{dom}_\varphi} (u \cdot u' - \varphi(u'))$, which we note is a standard function that is readily available in closed form for many choices of $\varphi$.

**Theorem 1** *Let $\varphi : \mathbb{R} \to (-\infty, \infty]$ be a convex lower semicontinuous mapping such that $\varphi(1) = 0$. Let $f$ be measurable and bounded. For any $\varepsilon > 0$, it holds that*

$$\sup_{\mathbf{x} \in \mathcal{X}} \inf_{q \in B_\varphi^t(p)} \mathbb{E}_{c \sim q}[f(\mathbf{x}, c)] = \sup_{\mathbf{x} \in \mathcal{X}, \lambda \geq 0, b \in \mathbb{R}} \left( b - \lambda \varepsilon_t - \lambda \mathbb{E}_{p_t(c)} \left[ \varphi^\star \left( \frac{b - f(\mathbf{x}, c)}{\lambda} \right) \right] \right).$$

**Proof (Sketch)** The proof begins by rewriting the constraint over the $\varphi$-divergence constrained ball with the use of Lagrangian multipliers. Using existing identities for $f$-divergences, a minimax swap yields a two-dimensional optimization problem, over $\lambda \geq 0$ and $b \in \mathbb{R}$.

We remark that similar results exist for other areas such as supervised learning [50], robust optimization [4] and certifying robust radii [14]. However this is, to the best of our knowledge, the first development when applied to optimizing expensive black-box functions, the case of BO. The above Theorem is practically compelling for three main reasons. First, one can note that compared to the left-hand side, the result converts this into a single optimization (max) over three variables, where two of the variables are 1-dimensional, reducing the computational burden significantly. Secondly, the notoriously difficult max-min problem becomes only a max, leaving behind instabilities one would encounter with the former objective. Finally, the result makes very mild assumptions on the context parameter space $\mathcal{C}$, allowing infinite spaces to be chosen, which is one of the challenges for existing BO advancements. We show that for specific choices of $\varphi$, the optimization over $b$ and even $\lambda$ can be expressed in closed form and thus simplified. All proofs for the following examples can be found in the Appendix Section 8.

**Example 2 ($\chi^2$-divergence)** *Let $\varphi(u) = (u-1)^2$, then for any measurable and bounded $f$ we have for any choice of $\varepsilon_t$*

$$\sup_{\mathbf{x} \in \mathcal{X}} \inf_{q \in B_\varphi^t(p_t)} \mathbb{E}_{c \sim q}[f(\mathbf{x}, c)] = \sup_{\mathbf{x} \in \mathcal{X}} \left( \mathbb{E}_{p_t(c)}[f(\mathbf{x}, c)] - \sqrt{\varepsilon_t \cdot \text{Var}_{p_t(c)}[f(\mathbf{x}, c)]} \right).$$

The above example can be easily implemented as it involves the same optimization problem however now appended with a variance term. Furthermore, this objective admits a compelling conceptual insight which is that, by enforcing a penalty in the form of variance, one attains robustness. The idea that regularization provides guidance to robustness or generalization is well-founded in machine learning more generally for example in supervised learning [12, 13]. We remark that this penalty and its relationship to $\chi^2$-divergence has been developed in the similar yet related problem of Bayesian quadrature [33]. Moreover, it can be shown that if $\varphi$ is twice differentiable then $D_\varphi$ can be approximated by the $\chi^2$-divergence via Taylor series, which makes $\chi^2$-divergence a centrally appealing choice for studying robustness. We now derive the result for a popular choice of $\varphi$ that is not differentiable.

**Example 3 (Total Variation)** *Let $\varphi(u) = |u - 1|$, then for any measurable and bounded $f$ we have for any choice of $\varepsilon_t$*

$$\sup_{\mathbf{x} \in \mathcal{X}} \inf_{q \in B_\varphi^t(p_t)} \mathbb{E}_{c \sim q}[f(\mathbf{x}, c)] = \sup_{\mathbf{x} \in \mathcal{X}} \left( \mathbb{E}_{p_t(c)}[f(\mathbf{x}, c)] - \frac{\varepsilon_t}{2} \left( \sup_{c \in \mathcal{C}} f(\mathbf{x}, c) - \inf_{c \in \mathcal{C}} f(\mathbf{x}, c) \right) \right).$$

Similar to the $\chi^2$-case, the result here admits a variance-like term in the form of the difference between the maximal and minimal elements. We remark that such a result is conceptually interesting

since both losses admit an objective that resembles a mean-variance which is a natural concept in ML, but advocates for it from the perspective of distributional robustness. This result exists for the supervised learning in Duchi and Namkoong [11] however is completely novel for BO and also holds for a choice of non-differentiable $\varphi$, hinting at the deeper connection between $\varphi$-divergence DRO and variance regularization.

## 4.1 Optimization with the GP Surrogate

To handle the distributional robustness, we have rewritten the objective function using $\varphi$ divergences in Theorem 1. In DRBO setting, we sequentially select a next point $\mathbf{x}_t$ for querying a black-box function. Given the observed context $c_t \sim q$ coming from the environment, we evaluate the black-box function and observe the output as $y_t = f(\mathbf{x}_t, c_t) + \eta_t$ where the noise $\eta_t \sim \mathcal{N}(0, \sigma_f^2)$ and $\sigma_f^2$ is the noise variance.

As a common practice in BO, at the iteration $t$, we model the GP surrogate model using the observed data $\{\mathbf{x}_i, y_i\}_{i=1}^{t-1}$ and make a decision by maximizing the acquisition function which is build on top of the GP surrogate:

$$\mathbf{x}_t = \arg\max_{\mathbf{x} \in \mathcal{X}} \alpha(\mathbf{x}).$$

While our method is not restricted to the form of the acquisition function, for convenience in the theoretical analysis, we follow the GP-UCB [52]. Given the GP predictive

---
**Algorithm 1** DRBO with $\varphi$-divergence
---
1: **Input:** Max iteration $T$, initial data $D_0$, $\eta$
2: **for** $t = 1, \ldots, T$ **do**
3:     Fit and estimate GP hyperparameter given $D_{t-1}$
4:     Select a next input $\mathbf{x}_t = \arg\max \alpha(\mathbf{x})$
5:     $\chi^2$-divergence: $\alpha(\mathbf{x}) := \alpha^{\chi^2}(\mathbf{x})$ from Eq. (4)
6:     Total Variation: $\alpha(\mathbf{x}) := \alpha^{TV}(\mathbf{x})$ from Eq. (5)
7:     Observe a context $c_t \sim q$
8:     Evaluate the black-box $y_t = f(\mathbf{x}_t, c_t) + \eta_t$
9:     Augment $D_t = D_{t-1} \cup (\mathbf{x}_t, c_t, y_t)$
10: **end for**

---

mean and variance from Eqs. (7,8), we have the acquisition function for the $\chi^2$ in Example 2 as follows:

$$\alpha^{\chi^2}(\mathbf{x}) := \frac{1}{|C|} \sum_c \left[ \mu_t(\mathbf{x}, c) + \sqrt{\beta_t} \sigma_t(\mathbf{x}, c) \right] - \sqrt{\frac{\varepsilon_t}{|C|} \sum_c \left( \mu_t(\mathbf{x}, c) - \bar{\mu}_t \right)^2} \tag{4}$$

where $\beta_t$ is a explore-exploit hyperparameter defined in Srinivas et al. [52], $\bar{\mu}_t = \frac{1}{|C|} \sum_c \mu_t(\mathbf{x}, c)$ and $c \sim q$ can be generated in a one dimensional space to approximate the expectation and the variance. In the experiment, we select $q$ as the uniform distribution, but it is not restricted to. Similarly, an acquisition function for Total Variation in Example 3 is written as

$$\alpha^{TV}(\mathbf{x}) := \frac{1}{|C|} \sum_c \left[ \mu_t(\mathbf{x}, c) + \sqrt{\beta_t} \sigma_t(\mathbf{x}, c) \right] - \frac{\varepsilon_t}{2} \left( \max \mu_t(\mathbf{x}, c) - \min \mu_t(\mathbf{x}, c) \right). \tag{5}$$

We summarize all computational steps in Algorithm 1.

**Computational Efficiency against MMD.** We make an important remark that since we do not require our context space to be finite, our implementation scales only linearly with the number of context samples $|C|$ drawing from $q$. This allows us to discretize our space and draw as many context samples as required while only paying a linear price. On the other hand, the MMD [26] at every iteration of $t$ requires solving an $|C|$-dimensional constraint optimization problem that has no closed form solution. We refer to Section 5.2 for the empirical comparison.

## 4.2 Convergence Analysis

One of the main advantages of Kirschner et al. [26] is the choice of MMD makes the regret analysis simpler due to the nice structure and properties of MMD. In particular, the MMD is well-celebrated for a $O(t^{-1/2})$ convergence where no such results exist for $\varphi$-divergences. However, using Theorem 1, we can show a regret bound for the Total Variation with a simple boundedness assumption and show how one can extend this result to other $\varphi$-divergences. We begin by defining the *robust regret*, $R_T$, with $\varphi$-divergence balls:

$$R_T(\varphi) = \sum_{t=1}^{T} \inf_{q \in B_\varphi^t} \mathbb{E}_{q(c)}[f(\mathbf{x}_t^*, c)] - \inf_{q \in B_\varphi^t} \mathbb{E}_{q(c)}[f(\mathbf{x}_t, c)], \tag{6}$$

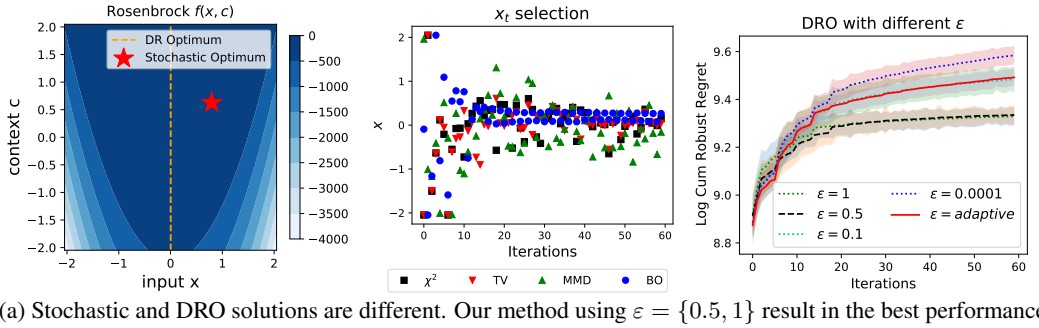

(a) Stochastic and DRO solutions are different. Our method using $\varepsilon = \{0.5, 1\}$ result in the best performance.

(b) Stochastic and DRO solutions are coincide. Our method with $\varepsilon \to 0$ is the best.

Figure 1: Two settings in DRO when the stochastic solution and robust solution are different (*top*) and identical (*bottom*). *Left*: original function $f(\mathbf{x}, c)$. *Middle*: selection of input $\mathbf{x}_t$ over iterations. *Right*: performance with different $\varepsilon$.

where $\mathbf{x}_t^* = \arg \max_{\mathbf{x} \in \mathcal{X}} \inf_{q \in B_{\varepsilon, \varphi}^t} \mathbb{E}_{q(c)}[f(\mathbf{x}, c)]$. We use $\boldsymbol{K}_t$ to denote the generated kernel matrix from dataset $D_t = \{(\mathbf{x}_i, c_i)\}_{i=1}^t \subset \mathcal{X} \times \mathcal{C}$. we now introduce a standard quantity in regret analysis in BO is the *maximum information gain*: $\gamma_t = \max_{D \subset \mathcal{X} \times \mathcal{C}: |D| = t} \log \det \left( \mathbf{I}_t + \sigma_f^{-2} \boldsymbol{K}_t \right)$ where $\boldsymbol{K}_t = [k([\mathbf{x}_i, c_i], [\mathbf{x}_j, c_j])]_{\forall i, j \leq t}$ is the covariance matrix and $\sigma_f^2$ is the output noise variance.

**Theorem 4 ($\varphi$-divergence Regret)** *Suppose the target function is bounded, meaning that $M = \sup_{(\mathbf{x}, c) \in \mathcal{X} \times \mathcal{C}} |f(\mathbf{x}, c)| < \infty$ and suppose $f$ has bounded RKHS norm with respect to $k$. For any lower semicontinuous convex $\varphi : \mathbb{R} \to (-\infty, \infty]$ with $\varphi(1) = 0$, if there exists a monotonic invertible function $\Gamma_\varphi : [0, \infty) \to \mathbb{R}$ such that $\mathrm{TV}(p, q) \leq \Gamma_\varphi(\mathsf{D}_\varphi(p, q))$, the following holds*

$$R_T(\varphi) \leq \frac{\sqrt{8 T \beta_T \gamma_T}}{\log(1 + \sigma_f^{-2})} + \left( 2M + \sqrt{\beta_T} \right) \sum_{t=1}^T \Gamma_\varphi(\varepsilon_t),$$

*with probability $1 - \delta$, where $\beta_t = 2\|f\|_k^2 + 300 \gamma_t \ln^3(t/\delta)$, $\gamma_t$ is the maximum information gain as defined above, and $\sigma_f$ is the standard deviation of the output noise.*

The full proof can be found in the Appendix Section 8. We first remark that with regularity assumptions on $f$, sublinear analytical bounds for $\gamma_T$ are known for a range of kernels, e.g., given $\mathcal{X} \times \mathcal{C} \subset \mathbb{R}^{d+1}$ we have for the RBF kernel, $\gamma_T \leq \mathcal{O}\left( \log(T)^{d+2} \right)$ or for the Matérn kernel with $\nu > 1$, $\gamma_T \leq \mathcal{O}\left( T^{\frac{(d+1)(d+2)}{2\nu + (d+1)(d+2)}} (\log T) \right)$. The second term in the bound is directly a consequence of DRO and by selecting $\varepsilon_t = 0$, it will vanish since any such $\Gamma_\varphi$ will satisfy $\Gamma_\varphi(0) = 0$. To ensure sublinear regret, we can select $\varepsilon_t = \Gamma_\varphi^{-1}\left( \frac{1}{\sqrt{t} + \sqrt{t+1}} \right)$, noting that the second term will reduce to $\sum_{t=1}^T \varepsilon_t \leq \sqrt{T}$. Finally, we remark that the existence of $\Gamma_\varphi$ is not so stringent since for a wide choices of $\varphi$, one can find inequalities between the Total Variation and $D_\varphi$, to which we refer the reader to Sason and Verdú [45]. For the examples discussed above, we can select $\Gamma_\varphi(t) = t$ for the TV. For the $\chi^2$ and KL cases, one can choose $\Gamma_{\chi^2}(b) = 2\sqrt{\frac{b}{1+b}}$ and $\Gamma_{\mathrm{KL}}(b) = 1 - \exp(-b)$.

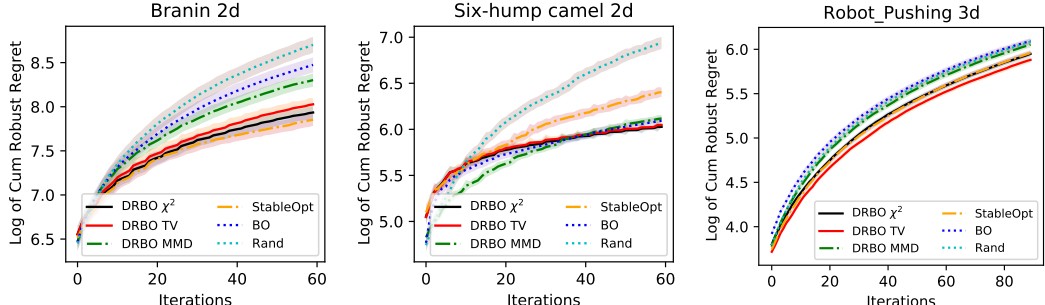

Figure 3: Cumulative robust regret across algorithms. The results show that the proposed $\chi^2$ and TV achieve the best performance across benchmark functions. Random and vanilla BO approaches perform poorly which do not take into account the robustness criteria. Best viewed in color.

# 5    Experiments

**Experimental setting.** The experiments are repeated using 30 independent runs. We set $|C| = 30$ which should be sufficient to draw $c \overset{\text{iid}}{\sim} q$ in one-dimensional space to compute Eqs. (4,5). We optimize the GP hyperparameter (e.g., learning rate) by maximizing the GP log marginal likelihood [43]. We will release the Python implementation code in the final version.

**Baselines.** We consider the following baselines for comparisons. *Rand*: we randomly select $\mathbf{x}_t$ irrespective of $c_t$. *BO*: we follow the GP-UCB [52] to perform standard Bayesian optimization (ignoring the context $c_t$). The selection at each iteration is $\mathbf{x}_t = \arg\max_{\mathbf{x}} \mu(\mathbf{x}) + \beta_t \sigma(\mathbf{x})$. *Stable-Opt*: we consider the worst-case robust optimization presented in Bogunovic et al. [8]. The selection at each iteration $\mathbf{x}_t = \arg\max_{\mathbf{x}} \arg\min_c \mu(\mathbf{x}, c) + \beta_t \sigma(\mathbf{x}, c)$. *DRBO MMD* [26]: Since there is no official implementation available, we have tried our best to re-implement the algorithm.

We consider the popular benchmark functions[3] with different dimensions $d$. To create a context variable $c$, we pick the last dimension of these functions to be the context input while the remaining $d - 1$ dimension becomes the input $\mathbf{x}$.

## 5.1    Ablation Studies

To gain understanding into how our framework works, we consider two popular settings below.

**DRBO solution is different from stochastic solution.** In Fig. 1a, the vanilla BO tends to converge greedily toward the stochastic solution (non-distributionally robust) $\arg\max_{\mathbf{x}} f(\mathbf{x}, \cdot)$. Thus, BO keeps exploiting in the locality of $\arg\max_{\mathbf{x}} f(\mathbf{x}, \cdot)$ from iteration 15. On the other hand, all other DRBO methods will keep exploring to seek for the distributionally robust solutions. Using the high value of $\varepsilon_t \in \{0.5, 1\}$ will result in the best performance.

**DRBO solution is identical to stochastic solution.** When the stochastic and robust solutions coincide at the same input $\mathbf{x}^*$, the solution of BO will be equivalent to the solution of DRBO methods. This is demonstrated by Fig. 1b. Both stochastic and robust approaches will quickly identify the optimal solution (see the $\mathbf{x}_t$ selection). We learn empirically that setting $\varepsilon_t \to 0$ will lead to the best performance. This is because the DRBO setting will become the standard BO.

The best choice of $\varepsilon$ depends on the property of the underlying function, e.g., the gap between the stochastic and DRBO solutions. In practice, we may not be able to identify these scenarios in advance. Therefore, we can use the adaptive value of $\varepsilon_t$ presented in Section 4.2. Using this adaptive setting, the performance is stable, as illustrated in the figures.

## 5.2    Computational efficiency

The key benefit of our framework is simplifying the existing intractable computation by providing the closed-form solution. Additional to improving the quality, we demonstrate this advantage in terms of computational complexity. Our main baseline for comparison is the MMD [26]. As shown in Fig. 2, our DRBO is consistently faster than the constraints linear programming approximation used for

---

[3]https://www.sfu.ca/ ssurjano/optimization.html

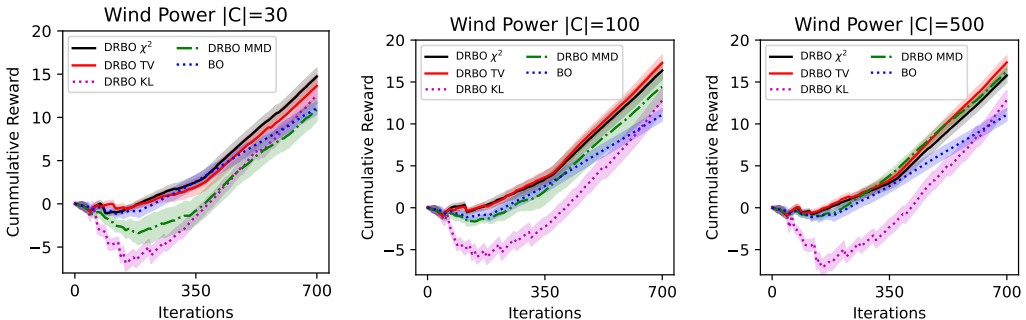

Figure 4: All divergences improve with larger $|C|$. However, MMD comes with the quadratic cost.

MMD. This gap is substantial in higher dimensions. In particular, as compared to Kirschner et al. [26], our DRBO is 5-times faster in *5d* and 10-times faster in *6d*.

### 5.3 Optimization performance comparison

We compare the algorithms in Fig. 3 using the robust (cumulative) regret defined in Eq. (6) which is commonly used in DRO literature [26, 33]. The random approach does not make any intelligent information in making decision, thus performs the worst. While BO performs better than random, it is still inferior comparing to other distributionally robust optimization approaches. The reason is that BO does not take into account the context information in making the decision. The StableOpt [8] performs relatively well that considers the worst scenarios in the subset of predefined context. This

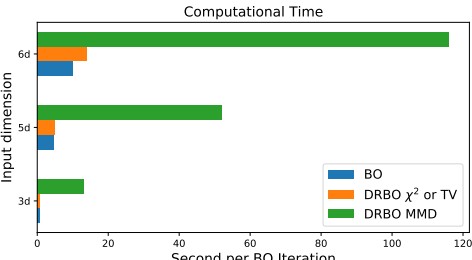

Figure 2: We compare the computational cost across methods. Our proposed DRBO using $\chi^2$ and TV take similar cost per iteration which is significantly lower than the DRBO MMD [26].

predefined subset can not cover all possible cases as opposed to the distributional robustness setting.

The MMD approach [26] needs to solve the inner adversary problem using linear programming with convex constraints, additional to the main optimization step. As a result, the performance of MMD is not as strong as our TV and $\chi^2$. Our proposed approach does not suffer this pathology and thus scale well in continuous and high dimensional settings of context input $c$.

**Real-world functions.** We consider the deterministic version of the robot pushing objective from Wang and Jegelka [60]. The goal is to find a good pre-image for pushing an object to a target location. The 3-dimensional function takes as input the robot location $(r_x, r_y) \in [-5, 5]^2$ and pushing duration $r_t \in [1, 30]$. We follow Bogunovic et al. [8] to twist this problem in which there is uncertainty regarding the precise target location, so one seeks a set of input parameters that is robust against a number of different potential pushing duration which is a context.

We perform an experiment on Wind Power dataset [8] and vary the context dimensions $|C| \in \{30, 100, 500\}$ in Fig. 4. When $|C|$ enlarges, our DRBO $\chi^2$, TV and KL improves. However, the performances do not improve further when increasing $|C|$ from 100 to 500. Similarly, MMD improves with $|C|$, but it comes with the quadratic cost w.r.t. $|C|$. Overall, our proposed DRBO still performs favourably in terms of optimization quality and computational cost than the MMD.

## 6 Conclusions, Limitations and Future works

In this work, we showed how one can study the DRBO formulation with respect to $\varphi$-divergences and derived a new algorithm that removes much of the computational burden, along with a sublinear regret bound. We compared the performance of our method against others, and showed that our results unveil a deeper connection between regularization and robustness, which serves useful conceptually.

**Limitations and Future Works** One of the limitations of our framework is in the choice of $\varphi$, for which we provide no guidance. For different applications, different choices of $\varphi$ would prove to be more useful, the study of which we leave for future work.

## Acknowledgements

We would like to thank the anonymous reviewers for providing feedback.

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
