**Gaussian Process with Input $x$ and Context $c$.** We have presented the definition of Gaussian process where the input includes a variable $\mathbf{x} \in \mathbb{R}^d$. Given the additional context variable $c \in \mathbb{R}$, it is natural to consider a GP model in which the input is the concatenation of $[\mathbf{x}, c] \in \mathbb{R}^{d+1}$. In particular, we can write a GP [43] as:

$$f([\mathbf{x}, c]) \sim \mathrm{GP}\left(m([\mathbf{x}, c]), k([\mathbf{x}, c], [\mathbf{x}', c'])\right),$$

where $m([\mathbf{x}, c])$ and $k([\mathbf{x}, c], [\mathbf{x}', c'])$ are the mean and covariance functions. For predicting $f_* = f([\mathbf{x}_*, c_*])$ at a new data point $[\mathbf{x}_*, c_*]$, the conditional probability follows a univariate Gaussian distribution as $p\left(f_* \mid [\mathbf{x}_*, c_*], ...\right) \sim \mathcal{N}\left(\mu([\mathbf{x}_*, c_*]), \sigma^2([\mathbf{x}_*, c_*])\right)$. The mean and variance are given by:

$$\mu([\mathbf{x}_*, c_*]) = \mathbf{k}_{*,N} \mathbf{K}_{N,N}^{-1} \mathbf{y}, \qquad (9) \qquad \sigma^2([\mathbf{x}_*, c_*]) = k_{**} - \mathbf{k}_{*,N} \mathbf{K}_{N,N}^{-1} \mathbf{k}_{*,N}^T \qquad (10)$$

where $k_{**} = k([\mathbf{x}_*, c_*], [\mathbf{x}_*, c_*])$, $\mathbf{k}_{*,N} = [k([\mathbf{x}_*, c_*], [\mathbf{x}_i, c_i])]_{\forall i \leq N}$ and $\boldsymbol{K}_{N,N} = [k([\mathbf{x}_i, c_i], [\mathbf{x}_j, c_j])]_{\forall i,j \leq N}$.

# 8 Proofs of Main Results

In the sequel, when we say a function is measurable, we attribute it with respect to the Borel $\sigma$-algebras based on the Polish topologies. We remark that a similar proof to ours has appeared in Ahmadi-Javid [2], which is not specific to BO objective yet also we require compactness of the set $\mathcal{C}$. We only require compactness of $\mathcal{C}$ to get compactness of $\Delta(\mathcal{C})$ however similar arguments can be made when $\mathcal{C}$ is not compact using the vague topology such as in [31, 24, 23].

**Theorem 5** *(Theorem 1 in the main paper)* *Let* $\varphi : \mathbb{R} \to (-\infty, \infty]$ *be a convex lower semicontinuous mapping such that* $\varphi(1) = 0$. *For any* $\varepsilon > 0$, *it holds that*

$$\sup_{\mathbf{x} \in \mathcal{X}} \inf_{q \in B_\varphi^t(p)} \mathbb{E}_{q(c)}[f(\mathbf{x}, c)] = \sup_{\mathbf{x} \in \mathcal{X}, \lambda \geq 0, b \in \mathbb{R}} \left( b - \lambda \varepsilon_t - \lambda \mathbb{E}_{p_t(c)} \left[ \varphi^\star \left( \frac{b - f(\mathbf{x}, c)}{\lambda} \right) \right] \right).$$

**Proof** For a fixed $\mathbf{x} \in \mathcal{X}$, we first introduce a Lagrangian variable $\lambda \geq 0$ that acts to enforce the ball constraint $\mathsf{D}_\varphi(p, q) \leq \varepsilon$:

$$\inf_{q \in B_\varphi^t(p)} \mathbb{E}_{q(c)}[f(\mathbf{x}, c)] = \inf_{q \in \Delta(\mathcal{C})} \sup_{\lambda \geq 0} \left( \mathbb{E}_{q(c)}[f(\mathbf{x}, c)] - \lambda \left( \varepsilon_t - \mathsf{D}_\varphi(q, p_t) \right) \right) \tag{11}$$

$$\overset{(1)}{=} \sup_{\lambda \geq 0} \inf_{q \in \Delta(\mathcal{C})} \left( \mathbb{E}_{q(c)}[f(\mathbf{x}, c)] - \lambda \left( \varepsilon_t - \mathsf{D}_\varphi(q, p_t) \right) \right) \tag{12}$$

$$= \sup_{\lambda \geq 0} \left( -\lambda \varepsilon_t - \sup_{q \in \Delta(\mathcal{C})} \left( \mathbb{E}_{q(c)}[-f(\mathbf{x}, c)] - \lambda \mathsf{D}_\varphi(q, p_t) \right) \right) \tag{13}$$

$$\overset{(2)}{=} \sup_{\lambda \geq 0} \left( -\lambda \varepsilon_t - \inf_{b \in \mathbb{R}} \left( \lambda \mathbb{E}_{p_t(c)} \left[ \varphi^\star \left( \frac{b - f(\mathbf{x}, c)}{\lambda} \right) \right] - b \right) \right) \tag{14}$$

$$= \sup_{\lambda \geq 0, b \in \mathbb{R}} \left( b - \lambda \varepsilon_t - \lambda \mathbb{E}_{p_t(c)} \left[ \varphi^\star \left( \frac{b - f(\mathbf{x}, c)}{\lambda} \right) \right] \right) \tag{15}$$

(1) is due to Fan's minimax Theorem [15, Theorem 2] noting that for any $\mathbf{x} \in \mathcal{X}$, the mapping $q \mapsto \mathbb{E}_{q(c)}[f(\mathbf{x}, c)]$ is linear and for any $\varphi$ chosen as stated in the Theorem, the mapping $q \mapsto \mathsf{D}_\varphi(q, p_t)$ is convex and lower semi-continuous. Furthermore noting that $\mathcal{C}$ is compact, we also have that $\Delta(\mathcal{C})$ is compact [58]. (2) is due to a standard result due to the (restricted) Fenchel dual of the $\varphi$-divergence, see Eq. (22) of Liu and Chaudhuri [31] for example. We state the result in the following lemma, which depends on the convex conjugate $\varphi^\star(u) = \sup_{u'} (uu' - \varphi(u'))$.

**Lemma 6** *For any measurable function $h : \mathcal{C} \to \mathbb{R}$ and convex lower semi-continuous function $\varphi : \mathbb{R} \to (-\infty, \infty]$ with $\varphi(1) = 0$, it holds that*

$$\sup_{q \in \Delta(\mathcal{C})} \left( \mathbb{E}_{q(c)}[h(c)] - \lambda \mathsf{D}_\varphi(q, p) \right) = \inf_{b \in \mathbb{R}} \left( \lambda \mathbb{E}_{p_t(c)} \left[ \varphi^\star \left( \frac{b + h(c)}{\lambda} \right) \right] - b \right),$$

*where $p \in \Delta(\mathcal{C})$ and $\lambda > 0$.*

∎

For each of the specific derivations below, we recall the standard derivations for $\varphi^\star$, which can be found in many works, for example in [38].

**Example 7** ($\chi^2$-**divergence**) *(Example 2 in the main paper) If $\varphi(u) = (u - 1)^2$, then we have*

$$\sup_{\mathbf{x} \in \mathcal{X}} \inf_{q \in B_\varphi^t(p)} \mathbb{E}_{c \sim q}[f(\mathbf{x}, c)] = \sup_{\mathbf{x} \in \mathcal{X}} \left( \mathbb{E}_{p_t(c)}[f(\mathbf{x}, c)] - \sqrt{\varepsilon_t \operatorname{Var}_{p_t(c)}[f(\mathbf{x}, c)]} \right).$$

**Proof** In this case we have $(\lambda\varphi)^\star(u) = \frac{u^2}{4\lambda} + u$ and so we have

$$\sup_{b \in \mathbb{R}} \left( b - \mathbb{E}_{p_t(c)} \left[ \frac{(b - f(\mathbf{x}, c))^2}{4\lambda} + b - f(\mathbf{x}, c) \right] \right) = \sup_{b \in \mathbb{R}} \left( \mathbb{E}_{p_t(c)}[f(\mathbf{x}, c)] - \mathbb{E}_{p_t(c)} \left[ (b - f(\mathbf{x}, c))^2 \right] \right)$$

$$= \mathbb{E}_{p_t(c)}[f(\mathbf{x}, c)] - \frac{1}{4\lambda} \inf_{b \in \mathbb{R}} \mathbb{E}_{p_t(c)} \left[ (b - f(\mathbf{x}, c))^2 \right]$$

$$= \mathbb{E}_{p_t(c)}[f(\mathbf{x}, c)] - \frac{1}{4\lambda} \operatorname{Var}_{p_t(c)}[f(\mathbf{x}, c)].$$

Combining this with the original objective and by Theorem 1 yields

$$\inf_{q \in B_\varphi^t(p)} \mathbb{E}_{c \sim q}[f(\mathbf{x}, c)] = \sup_{\lambda \geq 0, b \in \mathbb{R}} \left( b - \lambda \varepsilon_t - \lambda \mathbb{E}_{p_t(c)} \left[ \varphi^\star \left( \frac{b - f(\mathbf{x}, c)}{\lambda} \right) \right] \right)$$

$$= \mathbb{E}_{p_t(c)}[f(\mathbf{x}, c)] - \inf_{\lambda \geq 0} \left( \lambda \varepsilon_t + \frac{1}{4\lambda} \operatorname{Var}_{p_t(c)}[f(\mathbf{x}, c)] \right)$$

$$= \mathbb{E}_{p_t(c)}[f(\mathbf{x}, c)] - \sqrt{\varepsilon_t \operatorname{Var}_{p_t(c)}[f(\mathbf{x}, c)]}.$$

where the last equation is using the arithmetic and geometric means inequality. ∎

**Example 8 (Total Variation)** *(Example 3 in the main paper)* *If $\varphi(u) = |u - 1|$, then we have*

$$\sup_{\mathbf{x}\in\mathcal{X}} \inf_{q\in B_\varphi^t(p)} \mathbb{E}_{c\sim q}[f(\mathbf{x}, c)] = \sup_{\mathbf{x}\in\mathcal{X}} \left( \mathbb{E}_{p(c)}[f(\mathbf{x}, c)] - \frac{\varepsilon_t}{2} \left( \sup_{c\in\mathcal{C}} f(\mathbf{x}, c) - \inf_{c\in\mathcal{C}} f(\mathbf{x}, c) \right) \right).$$

**Proof** In this case, the conjugate of $\varphi$ is $(\lambda\varphi)^\star(u) = 0$ if $|u| \leq \lambda$ and $+\infty$ otherwise. Therefore, when considering the right-hand side of Theorem 1, we will require $\lambda$ to be larger than $|b - f(\mathbf{x}, c)|$ for all $c \in \mathcal{C}$ for the expression to be finite. In this case, for a fixed $b \in \mathbb{R}$, the optimization over $\lambda \geq 0$ becomes

$$\sup_{b\in\mathbb{R}} \sup_{\lambda\geq 0} \left( b - \lambda\varepsilon_t - \lambda\mathbb{E}_{p_t(c)} \left[ \varphi^\star \left( \frac{b - f(\mathbf{x}, c)}{\lambda} \right) \right] \right) \tag{16}$$

$$= \sup_{b\in\mathbb{R}} \left( \mathbb{E}_{p_t(c)}[f(\mathbf{x}, c)] - \sup_{c\in\mathcal{C}} |b - f(\mathbf{x}, c)| \, \varepsilon_t \right) \tag{17}$$

$$= \mathbb{E}_{p_t(c)}[f(\mathbf{x}, c)] - \inf_{b\in\mathbb{R}} \sup_{c\in\mathcal{C}} |b - f(\mathbf{x}, c)| \, \varepsilon_t. \tag{18}$$

We need the following Lemma to proceed.

**Lemma 9** *For any set of numbers $A \subset \mathbb{R}$, let $\overline{a}, \underline{a} \in A$ be the maximum and minimal elements. It then holds that*

$$\inf_{b\in\mathbb{R}} \sup_{a\in A} |b - a| = \frac{1}{2} |\overline{a} - \underline{a}|.$$

**Proof** First note that for any $b \in \mathbb{R}$, we have that $\sup_{a\in A} |b - a| = \max(|\overline{a} - b|, |\underline{a} - b|)$. The outer inf can be solved by setting $b = \frac{\overline{a} + \underline{a}}{2}$. ■

The proof then concludes by noting setting $A = \{f(\mathbf{x}, c) : c \in \mathcal{C}\}$. ■

**Lemma 10 (Theorem 6 from Srinivas et al. [52])** *Let $\delta \in (0, 1)$. Assume the noise variable $\varepsilon_t$ is uniformly bounded by $\sigma_f$. Define $\beta_t = 2||f||_k^2 + 300\gamma_t \ln^3(t/\delta)$. Then,*

$$P\left( \forall T, \forall \mathbf{x}, |\mu_T(\mathbf{x}) - f(\mathbf{x})| \leq \beta_T^{\frac{1}{2}} \sigma_T(\mathbf{x}) \right) \geq 1 - \delta. \tag{19}$$

We first prove a regret bound for the total variation which will be instrumental in proving the regret bound for any general $\varphi$, given the existence of $\Gamma_\varphi$.

**Theorem 11 (Total Variation Regret)** *Let $M = \sup_{(\mathbf{x},c)\in\mathcal{X}\times\mathcal{C}} |f(\mathbf{x}, c)|$ and suppose $f$ lives in an RKHS. If $\varphi(u) = |u - 1|$, it then holds that*

$$R_T(\varphi) \leq \frac{\sqrt{8T\beta_T\gamma_T}}{\log(1 + \sigma_f^{-2})} + \left( 2M + \sqrt{\beta_T} \right) \sum_{t=1}^{T} \varepsilon_t,$$

*where $\sigma_f$ is the standard deviation of the output noise.*

**Proof** We first define the GP predictive mean and variance as

$$\mu_t(\mathbf{x}, c) = k_t([\mathbf{x}, c])^\intercal \left( \boldsymbol{K}_t + \mathbf{I}_t\sigma_f^2 \right)^{-1} y_t$$

$$\sigma_t(\mathbf{x}, c)^2 = k([\mathbf{x}, c], [\mathbf{x}, c]) - k_t([\mathbf{x}, c])^\intercal \left( \boldsymbol{K}_t + \mathbf{I}_t\sigma_f^2 \right)^{-1} k_t([\mathbf{x}, c]),$$

which are defined in Eqs. (9,10) where $\boldsymbol{K}_t = [k([\mathbf{x}_i, c_i], [\mathbf{x}_j, c_j])]_{\forall i,j\leq t}$. The proof begins by first bounding the regret at time $t$ using a standard argument with a slight modification that uses our

results.

$$r_t = \inf_{q:\mathsf{D}_\varphi(q,p_t)\leq\varepsilon_t} \mathbb{E}_{q(c)}[f(\mathbf{x}^*,c)] - \inf_{q:\mathsf{D}_\varphi(q,p_t)\leq\varepsilon_t} \mathbb{E}_{q(c)}[f(\mathbf{x}_t,c)]$$

$$\overset{(1)}{\leq} \mathbb{E}_{p_t(c)}[f(\mathbf{x}^*,c) - \mu(\mathbf{x}^*,c)] + \mathbb{E}_{p_t(c)}[\mu(\mathbf{x}^*,c)] - \inf_{q:\mathsf{D}_\varphi(q,p_t)\leq\varepsilon_t} \mathbb{E}_{q(c)}[f(\mathbf{x}_t,c)]$$

$$\overset{(2)}{\leq} \sqrt{\beta_t}\mathbb{E}_{p_t(c)}[\sigma_t(\mathbf{x}^*,c)]] + \mathbb{E}_{p_t(c)}[\mu(\mathbf{x}^*,c)] - \inf_{q:\mathsf{D}_\varphi(q,p_t)\leq\varepsilon_t} \mathbb{E}_{q(c)}[f(\mathbf{x}_t,c)]$$

$$= \sqrt{\beta_t}\mathbb{E}_{p_t(c)}[\sigma_t(\mathbf{x}^*,c)] + \mathbb{E}_{p_t(c)}[\mu(\mathbf{x}^*,c)] - \inf_{q:\mathsf{D}_\varphi(q,p_t)\leq\varepsilon_t} \mathbb{E}_{q(c)}[f(\mathbf{x}_t,c)]$$

$$\overset{(3)}{=} \sqrt{\beta_t}\mathbb{E}_{p_t(c)}[\sigma_t(\mathbf{x}^*,c)] + \mathbb{E}_{p_t(c)}[\mu(\mathbf{x}^*,c)] - \mathbb{E}_{p_t(c)}[f(\mathbf{x}_t,c)] + \frac{\varepsilon_t}{2}\left(\sup_{c\in\mathcal{C}} f(\mathbf{x}_t,c) - \inf_{c\in\mathcal{C}} f(\mathbf{x}_t,c)\right)$$

$$= \sqrt{\beta_t}\mathbb{E}_{p_t(c)}[\sigma_t(\mathbf{x}^*,c)] + \mathbb{E}_{p_t(c)}[\mu(\mathbf{x}^*,c) - f(\mathbf{x}_t,c)] + \frac{\varepsilon_t}{2}\left(\sup_{c\in\mathcal{C}} f(\mathbf{x}_t,c) - \inf_{c\in\mathcal{C}} f(\mathbf{x}_t,c)\right)$$

$$\overset{(4)}{\leq} \sqrt{\beta_t}\mathbb{E}_{p_t(c)}[\sigma_t(\mathbf{x}_t,c)] + \mathbb{E}_{p_t(c)}[\mu(\mathbf{x}_t,c) - f(\mathbf{x}_t,c)] + \frac{\varepsilon_t}{2}\left(\sup_{c\in\mathcal{C}} f(\mathbf{x}_t,c) - \inf_{c\in\mathcal{C}} f(\mathbf{x}_t,c)\right)$$

$$+ \frac{\varepsilon_t}{2}\mathbb{E}_{p_t(c)}\left[\max \mu_t(\mathbf{x}^*,c) - \min \mu_t(\mathbf{x}^*,c)\right]$$

$$\overset{(5)}{\leq} 2\sqrt{\beta_t}\mathbb{E}_{p_t(c)}[\sigma_t(\mathbf{x}_t,c)] + \varepsilon_t\big(2M + \sqrt{\beta_t}\big), \tag{20}$$

where (1) holds due to selecting $p_t$ and introducing $\mathbb{E}_{q(c)}[\mu(\mathbf{x}^*,c)]$, (2) is due to the result that $|f(\mathbf{x},c) - \mu(\mathbf{x},c)| \leq \sqrt{\beta_t}\sigma_t(\mathbf{x},c)$ for all $x,c \in \mathcal{X} \times \mathcal{C}$ with probability at least $1-\delta$ as stated in Lemma 10. (3) is due to Theorem 8. Step (4) is due to the choice of $\mathbf{x}_t$ since it satisfies:

$$\mathbb{E}_{p_t(c)}\left[\mu(\mathbf{x}_t,c) + \sqrt{\beta_t}\sigma_t(\mathbf{x}_t,c) - \frac{\varepsilon_t}{2}\big(\max \mu_t(\mathbf{x}_t,c) - \min \mu_t(\mathbf{x}_t,c)\big)\right] \geq$$
$$\mathbb{E}_{p_t(c)}\left[\mu(\mathbf{x},c) + \sqrt{\beta_t}\sigma_t(\mathbf{x},c) - \frac{\varepsilon_t}{2}\big(\max \mu_t(\mathbf{x},c) - \min \mu_t(\mathbf{x},c)\big)\right],$$

for all $\mathbf{x} \in \mathcal{X}$ and therefore

$$\mathbb{E}_{p_t(c)}\left[\mu(\mathbf{x}_t,c) + \sqrt{\beta_t}\sigma_t(\mathbf{x}_t,c) - \frac{\varepsilon_t}{2}\big(\max \mu_t(\mathbf{x}_t,c) - \min \mu_t(\mathbf{x}_t,c)\big)\right] \geq$$
$$\mathbb{E}_{p_t(c)}\left[\mu(\mathbf{x}^*,c) + \sqrt{\beta_t}\sigma_t(\mathbf{x}^*,c) - \frac{\varepsilon_t}{2}\big(\max \mu_t(\mathbf{x}^*,c) - \min \mu_t(\mathbf{x}^*,c)\big)\right].$$

Thus, we have step (4) as

$$\mathbb{E}_{p_t(c)}[\mu(\mathbf{x}^*,c)] + \sqrt{\beta_t}\mathbb{E}_{p_t(c)}[\sigma_t(\mathbf{x}^*,c)] \leq \mathbb{E}_{p_t(c)}[\mu(\mathbf{x}_t,c)] + \sqrt{\beta_t}\mathbb{E}_{p_t(c)}[\sigma_t(\mathbf{x}_t,c)]$$
$$+ \frac{\varepsilon_t}{2}\mathbb{E}_{p_t(c)}\left[\max \mu_t(\mathbf{x}^*,c) - \min \mu_t(\mathbf{x}^*,c)\right].$$

Finally, (5) holds due to another application of $|f(\mathbf{x},c) - \mu(\mathbf{x},c)| \leq \beta_t\sigma_t(\mathbf{x},c)$ for all $[\mathbf{x},c] \in [\mathcal{X} \times \mathcal{C}]$ from Lemma 10, $\mathbb{E}_{p_t(c)}\left[\sup_{c\in\mathcal{C}} f(\mathbf{x}_t,c) - \inf_{c\in\mathcal{C}} f(\mathbf{x}_t,c)\right] \leq 2M$, and $\mathbb{E}_{p_t(c)}\left[\max \mu_t(\mathbf{x}^*,c) - \min \mu_t(\mathbf{x}^*,c)\right] \leq 2[M + \sqrt{\beta_t}\sigma_t(\mathbf{x}^*,c)] \leq 2[M + \sqrt{\beta_t}]$ where $M = \sup_{(\mathbf{x},c)\in\mathcal{X}\times\mathcal{C}} |f(\mathbf{x},c)|$ and $\sigma_t(\mathbf{x},c) \leq 1, \forall[\mathbf{x},c]$. For the final step of our proof, we follow the literature in Bayesian optimization [52] to introduce a sample complexity parameter, namely the maximum information gain:

$$\gamma_T := \max_{\{(\mathbf{x}_t,c_t)\}_{t=1}^T} \log\det(\mathbf{I}_t + \sigma_f^{-2}\boldsymbol{K}_T).$$

where we use $\boldsymbol{K}_T$ to denote the generated kernel matrix from dataset $D_T = \{[\mathbf{x}_i,c_i]\}_{i=1}^T \subset [\mathcal{X} \times \mathcal{C}]$ at iteration $T$. The information gain is used in the regret bounds for most of Bayesian optimization research [35, 26].

**Lemma 12** *(adapted Lemma 7 in Nguyen et al. [35]) The sum of the predictive variances is bounded by the maximum information gain $\gamma_T$. That is $\forall \mathbf{x},c \in \mathcal{X} \times \mathcal{C}, \sum_{t=1}^T \sigma_{t-1}^2(\mathbf{x},c) \leq \frac{2}{\log(1+\sigma_f^{-2})}\gamma_T$ where $\sigma_f$ is the standard deviation of the output noise.*

Using the above Lemma of maximum information gain, we take the square of the term $2\sqrt{\beta_t}\mathbb{E}_{p_t(c)}[\sigma_t(\mathbf{x}_t, c)]$ in Eq. (20) to have:

$$\sum_{t=1}^{T} 4\beta_t \mathbb{E}_{p_t(c)}[\sigma_t^2(\mathbf{x}_t, c)] \leq \frac{8\beta_T \gamma_T}{\log(1 + \sigma_f^{-2})} \tag{21}$$

By using Cauchy-Schwarz inequality, we get

$$\sum_{t=1}^{T} 2\sqrt{\beta_t}\mathbb{E}_{p_t(c)}[\sigma_t(\mathbf{x}, c)] \leq \sqrt{\frac{8T\beta_T\gamma_T}{\log(1 + \sigma_f^{-2})}} \tag{22}$$

in which the term $T$ has been included in the right.

Using the above results we get

$$R_T(\varphi) \leq 2\sum_{t=1}^{T} \sqrt{\beta_t}\mathbb{E}_{p_t(c)}[\sigma_t(\mathbf{x}, c)] + \left(2M + \sqrt{\beta_T}\right)\sum_{t=1}^{T} \varepsilon_t \tag{23}$$

$$\leq 2\sqrt{\beta_T}\sum_{t=1}^{T} \max_c[\sigma_t(\mathbf{x}, c)] + \left(2M + \sqrt{\beta_T}\right)\sum_{t=1}^{T} \varepsilon_t \tag{24}$$

$$\leq 2\sqrt{\beta_T} \max_{q \in \{p_1, \ldots, p_T\}} \sum_{t=1}^{T} \mathbb{E}_{q(c)}[\sigma_t(\mathbf{x}, c)] + \left(2M + \sqrt{\beta_T}\right)\sum_{t=1}^{T} \varepsilon_t \tag{25}$$

$$\leq 2\sqrt{\beta_T} \max_{q \in \{p_1, \ldots, p_T\}} \mathbb{E}_{q(c)}\left[\sum_{t=1}^{T} \sigma_t(\mathbf{x}, c)\right] + \left(2M + \sqrt{\beta_T}\right)\sum_{t=1}^{T} \varepsilon_t \tag{26}$$

$$\leq \frac{\sqrt{8T\beta_T\gamma_T}}{\log(1 + \sigma_f^{-2})} + \left(2M + \sqrt{\beta_T}\right)\sum_{t=1}^{T} \varepsilon_t \tag{27}$$

where we have used Eq. (22) to obtained Eq. (27). ∎

**Lemma 13** *For any $T > 0$, it holds that*

$$\sum_{t=1}^{T} \left(\frac{1}{\sqrt{t} + \sqrt{t+1}}\right) = \sqrt{T+1} - 1 \leq \sqrt{T},$$

**Proof** By simply rationalizing the denominator, we have that

$$\frac{1}{\sqrt{t} + \sqrt{t+1}} = \sqrt{t+1} - \sqrt{t},$$

and via a simple telescoping sum, the required result holds. A simple inequality will then yield the final inequality. ∎

**Theorem 14 ($\varphi$-divergence Regret)** *(Theorem 4 in the main paper) Let $M = \sup_{(\mathbf{x},c) \in \mathcal{X} \times \mathcal{C}} |f(\mathbf{x}, c)| < \infty$ and suppose $f$ has bounded RKHS norm with respect to $k$. For any lower semicontinuous convex $\varphi : \mathbb{R} \to (-\infty, \infty]$ with $\varphi(1) = 0$, if there exists a monotonic invertible function $\Gamma_\varphi : [0, \infty) \to \mathbb{R}$ such that $\mathrm{TV}(p, q) \leq \Gamma_\varphi(\mathsf{D}_\varphi(p, q))$, the following holds*

$$R_T(\varphi) \leq \frac{\sqrt{8T\beta_T\gamma_T}}{\log(1 + \sigma_f^{-2})} + \left(2M + \sqrt{\beta_T}\right)\sum_{t=1}^{T} \Gamma_\varphi(\varepsilon_t),$$

*with probability $1 - \delta$, where $\beta_t = 2||f||_k^2 + 300\gamma_t \ln^3(t/\delta)$, $\gamma_t$ is the maximum information gain as defined above, and $\sigma_f$ is the standard deviation of the output noise.*

**Proof** The key aspect of this proof is to note that

$$\{q \in \Delta(\mathcal{C}) : \mathsf{D}_\varphi(q, p_t) \leq \varepsilon_t\} \subseteq \{q \in \Delta(\mathcal{C}) : \mathrm{TV}(q, p_t) \leq \Gamma_\varphi(\varepsilon_t)\}, \tag{28}$$

which is due to the inequality and monotonicity of the $\Gamma_\varphi$ function. By following similar steps to the Total Variation derivation, we have

$$\inf_{q:\mathsf{D}_\varphi(q,p_t)\leq\varepsilon_t} \mathbb{E}_{q(c)}[f(\mathbf{x}^*, c)] - \inf_{q:\mathsf{D}_\varphi(q,p_t)\leq\varepsilon_t} \mathbb{E}_{q(c)}[f(\mathbf{x}_t, c)] \tag{29}$$

$$\leq \sqrt{\beta_t}\mathbb{E}_{p_t(c)}[\sigma_t(\mathbf{x}^*, c)^2 + \mu(\mathbf{x}^*, c)] - \inf_{q:\mathsf{D}_\varphi(q,p_t)\leq\varepsilon_t} \mathbb{E}_{q(c)}[f(\mathbf{x}_t, c)] \tag{30}$$

$$\leq \sqrt{\beta_t}\mathbb{E}_{p_t(c)}[\sigma_t(\mathbf{x}^*, c)^2 + \mu(\mathbf{x}^*, c)] - \inf_{q:\mathrm{TV}(q,p_t)\leq\Gamma_\varphi(\varepsilon_t)} \mathbb{E}_{q(c)}[f(\mathbf{x}_t, c)]. \tag{31}$$

By following the same decomposition as in the proof for the Total Variation, presented in Theorem 11, we achieve the same result except with the radius epsilon replaced with $\Gamma_\varphi(\varepsilon_t)$. ∎

# 9 Extension to KL divergence

Our theoretical result for $\varphi$-divergence can also be readily extended to handle KL divergence. However, since the empirical result with KL divergence is inferior to the result with TV and $\chi$-divergences.

**Example 15 (KL-divergence)** *Let $\varphi(u) = u \log u$, we have*

$$\sup_{\mathbf{x}\in\mathcal{X}} \inf_{q\in B_\varphi^t(p)} \mathbb{E}_{c\sim q}[f(\mathbf{x}, c)] = \sup_{\mathbf{x}\in\mathcal{X}, \lambda\geq 0} \left(-\lambda\varepsilon_t - \lambda \log \mathbb{E}_{p_t(c)}\left[\exp\left(\frac{-f(\mathbf{x}, c)}{\lambda}\right)\right]\right).$$

**Proof** Using Theorem 1, we derive $\varphi^\star$ for this choice which can easily be verified to be $\varphi^\star(t) = \exp(t-1)$. For simplicity, let $\lambda = 1$ and note that we have

$$\sup_{b\in\mathbb{R}} \left(b - \mathbb{E}_{p_t(c)}\left[\exp\left(b - f(\mathbf{x}, c) - 1\right)\right]\right) = \sup_{b\in\mathbb{R}} \left(b - \exp\left(b\right) \cdot A\right), \tag{32}$$

where $A = \mathbb{E}_{p_t(c)}\left[\exp\left(-f(\mathbf{x}, c) - 1\right)\right]$ is a constant. The above can easily be solved as it admits a differentiable one-dimensional objective and we get the largest value when $b = -\log A$ which then yields

$$\sup_{b\in\mathbb{R}} \left(b - \mathbb{E}_{p_t(c)}\left[\exp\left(b - f(\mathbf{x}, c) - 1\right)\right]\right) = \log \mathbb{E}_{p_t(c)}\left[\exp(-f(\mathbf{x}, c))\right]. \tag{33}$$

The proof concludes noting that $(\lambda\varphi)^\star(u) = \lambda\varphi^\star(u/\lambda)$ for any $\lambda > 0$. ∎

The Kullback-Leibler (KL) divergence [28] is a popular choice when quantifying information shift, due to its link with entropy. There exists work that studied distributional shifts with respect to the KL divergence for label shifts [61]. Compared to the general theorem, the KL-divergence variant allows us to find $b \in \mathbb{R}$ in closed form. We remark that we place the KL divergence derivation here for its information-theoretic importance however in our experiments, we find that other choices of $\varphi$-divergence outperform the KL-divergence. We now show such examples and particularly illustrate that we can even solve for $\lambda \geq 0$ in closed form for these cases, yielding only a single maximization over $\mathcal{X}$.

The regret bound for the case with KL divergence is presented in Theorem 4 using $\Gamma_{\mathrm{KL}}(t) = 1 - \exp(-t)$.

## 9.1 Optimization with the GP Surrogate for KL-divergence

To handle the distributional robustness, we have rewritten the objective function using $\varphi$ divergences in Theorem 1 with the KL in **Example** 15.

Similar to the cases of TV and $\chi^2$, we model the GP surrogate model using the observed data $\{\mathbf{x}_i, y_i\}_{i=1}^{t-1}$ at each iteration $t$-th, . Then, we select a next point $\mathbf{x}_t$ to query a black-box function by maximizing the acquisition function which is build on top of the GP surrogate:

$$\mathbf{x}_t = \arg\max_{\mathbf{x}\in\mathcal{X}} \alpha(\mathbf{x}).$$

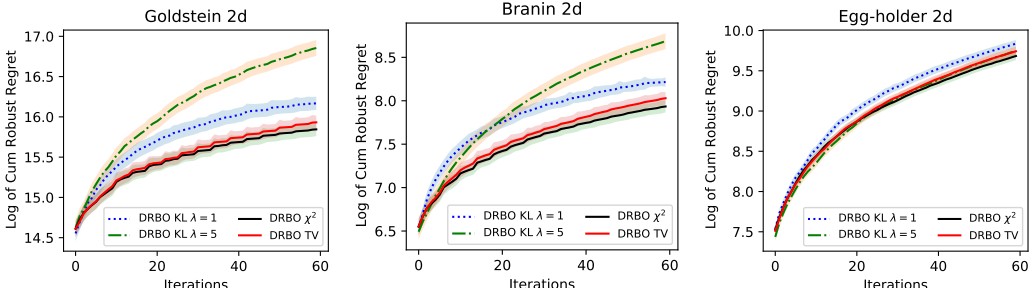

Figure 5: We empirically found that using TV and $\chi^2$ divergences typically obtain better performance than KL divergence. This can be due to the sensitivity of the additional hyperparameter $\lambda$ in KL. We have considered using $\lambda = 1$ and $\lambda = 5$ in these experiments. The best $\lambda$ is unknown in advance and depends on the functions.

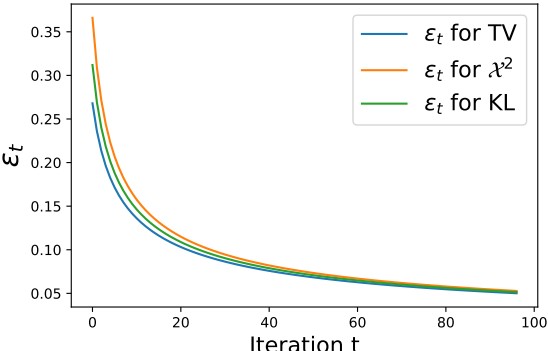

Figure 6: The adaptive value of $\varepsilon_t$ over iterations, $\lim_{t \to \infty}, \varepsilon_t = 0$.

While our method is not restricted to the form of the acquisition function, for convenient in the theoretical analysis, we follow the GP-UCB [52]. Given the GP predictive mean and variance from Eqs. (9,10), we have the acquisition function for the KL in Example 15 as follows:

$$\alpha^{KL}(\mathbf{x}) := -\lambda \varepsilon_t - \lambda \log \mathbb{E}_{p_t(c)} \left[ \exp \left( \frac{-\frac{1}{|C|} \sum_c \left[ \mu_t(\mathbf{x}, c) + \sqrt{\beta_t} \sigma_t(\mathbf{x}, c) \right]}{\lambda} \right) \right] \quad (34)$$

where $c \sim q$ can be generated in a one dimensional space to approximate the expectation and the variance. In the experiment, we define $q$ as the uniform distribution to draw $c \sim q$.

**Comparison of KL, TV and $\chi^2$ divergences.** In Fig. 5, we present the additional experiments showing the comparison of using different divergences including KL, TV and $\chi^2$. We show empirically that the KL divergence performs generally inferior than the TV and $\chi^2$.

**Adaptive value of $\varepsilon_t$.** We show the adaptive value of $\varepsilon_t$ by iterations for TV, $\chi^2$ and KL in Fig. 6.

In Fig. 7, we present additional visualization to complement the analysis in Fig. 1. In particular, we illustrate three other functions including branin, goldstein and six-hump camel. The additional results are consistent with the finding presented in the main paper (Section 5.1 and Fig. 1).

## 9.2 Selection radii for $\varphi$-divergences

Regarding the choice of radii $\varepsilon_t$, if we apply Gaussian smoothing on $p_t$ then there exists convergence rates for $\varphi$-divergences from Goldfeld et al. [17]. Therefore we can select this choice and ensure that the population distribution $P$ is within the DRO-BO ball. We emphasize that we can choose smoothed distributions since the main Theorem holds for continuous context distributions, a feature specific to our contribution.

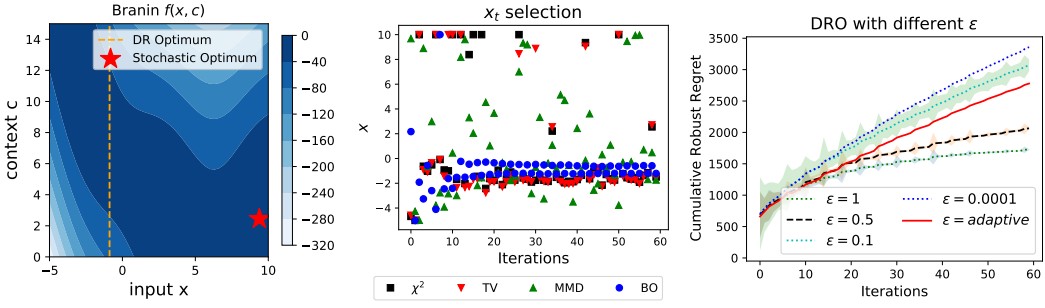

(a) Stochastic and DRO solution are different. The choices of $\varepsilon = \{0.5, 1\}$ result in the best performance.

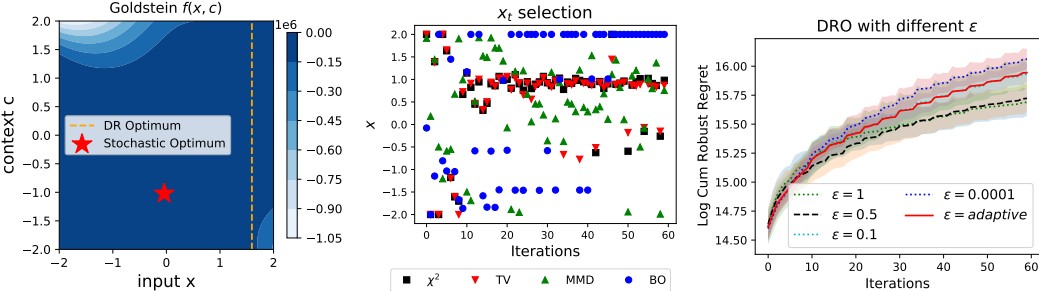

(b) Stochastic and DRO solution are different. The choices of $\varepsilon = \{0.5, 1\}$ result in the best performance.

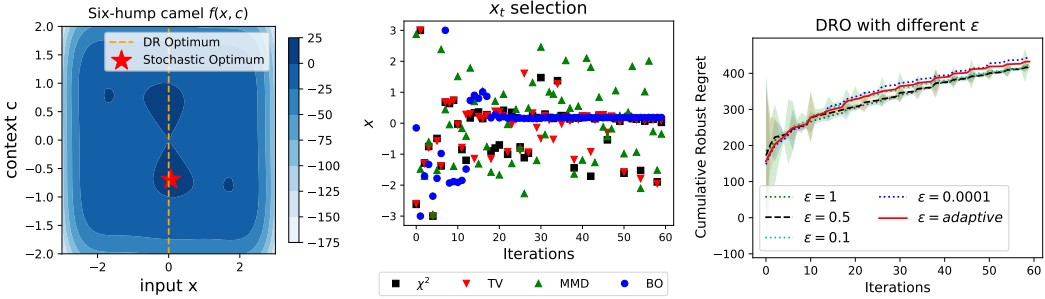

(c) Stochastic and DRO solution are coincide. $\varepsilon \to 0$ is the best.

Figure 7: We complement the result presented in Fig. 1 using three additional functions. There are two settings in DRO when the stochastic solution and robust solution are different (*top*) and identical (*bottom*). *Left*: the original function $f(\mathbf{x}, c)$. *Middle*: the selection of input $\mathbf{x}_t$ over iterations. *Right*: optimization performance with different $\varepsilon$. The adaptive choice of $\varepsilon_t$ (in red) always produces stable performance across various choices of $\varepsilon_t$. This is especially useful in unknown functions where we do not have prior assumption on the underlying structure to decide on which large or small values of $\varepsilon_t$ to be specified.