# OpenReview forum: "Distributionally Robust Bayesian Optimization with $\varphi$-divergences"
_NeurIPS.cc/2023/Conference — NeurIPS 2023 poster_

### Official Review · Reviewer_NSv3 · 2023-06-27

**Soundness:** 2 fair
**Presentation:** 2 fair
**Contribution:** 3 good
**Rating:** 3
**Confidence:** 4

**Summary:**

The authors extend the framework of distributionally robust Bayesian optimization to the case where the distribution distance notion amounts to $\phi$-divergences, which encompasses the Kullback-Leibler divergence, total variation and $\chi^2$-divergence. In particular, the paper aims at  providing a computationally tractable algorithm for the maximization of a reward function that additionally depends on a context parameter that is drawn from a distribution with respect to which the procedure is supposed to be distributionally robust.

They build on the paper [23] by Kirschner et al., where a similar problem was considered and where an efficient algorithm based on convex optimization was developed for the said problem if robustness is considered with respect to the maximum mean discrepancy distance.

The main result of the paper is Theorem 1, which allows to rewrite the maximization of the distributionally robust objective (DRO) as maximization of a standard stochastic optimization objective correct by a variance term in the cases of total variation and $\chi^2$-divergence. This is the result of a characterization of the infimum in the DRO using complex conjugates of the $\phi$-function. Furthermore, a robust regret bound is derived and some numerical experiments are conducted for simple, standard reference function such as the Rosenbrock and cosine functions.

**Strengths:**

The main strength of the paper lies from my view in the fact that the authors provide tractable characterizations of the distributionally robust objective (see page 4) if the distributional robustness is measured with respect to the KL, total variation and $\chi^2$-divergences. I consider the computations that lead to these characterizations to be technically sound.

This extends prior work by Kirschner et al. that considered only distributional robustness with respect to the maximum mean discrepancy.


**Weaknesses:**

A main weakness of the submission is that despite the claim that its results go beyond the finiteness assumption on the set of contexts  [23], this is not the case: In the definition of (4) and (5), the size of the set (which is called space) of contexts is implicitly assumed to be finite due to the summation over c and the division by the cardinality |C| of the context set.  This can be, for example, also seen in line 168 of page 4 where the finiteness assumption on the reference distribution is made explicit. A clarification or weakening of the claims that is necessary from my point of view.

For a through understanding of the bound on the robust regret presented in Theorem 2, a clear comparison to the bounds obtained in other papers such as [23] would be desirable.

The experiments that are considered appear rather consider very low-dimensional functions without a clear link to a machine learning related topic.

The presentation of the paper lacks some clarity and suffers from challenges in its use of the English language. Furthermore, at several occasions, notation is used that is not explained or only much later in the manuscript. We list a few issues below:
- Second sentence is not grammatically correct
- "computed, however cannot be replaced by another choice of D whose closed form is not readily accessible with samples such as the ϕ-divergence"
- "samples such as the $\phi$-divergence
- $\varepsilon$ without dash in line 70, page 2
- superfluous bracket in p. 3, line 119
- "include the Wasserstein". Probably some sort of Wasserstein distance is referred to here, but is so far unclear
- "Total Variance", line p.4, line 153
- The convex conjugate would profit from an appropriate evaluation
- "the same optimization problem" To which comparison optimization problem is this referencing?
- "that" missing on p.6, line 253
- In (4) and (5), a lot of quantities are not defined, including $\mu_t$, $\beta_t$ (in the mentioned reference [47], that parameter is not defined explicitly either) and $\sigma_t$.
- It is unclear what is needed exactly in Algorithm 1 to run the algorithm. Do we need to know the values of $\varepsilon_t$, the reference distribution p_t, the kernel?
- P. 3, line 106: $\mathcal{X} \in \mathbb{R}^d$ is probably not correct here



**Questions:**

- Why is the maximum information gain defined differently than in Kirschner et al.? Elaborate.
- Please rework the presentation of the manuscript to better clarify the underlying assumptions, missing definitions and to address grammar issues.
- The motivation of the comparisons of the different DRBOs in Figure 2 and Figure 3 is currently unclear: The different curves correspond to different settings (balls of distributions with respect to different distance notions). Why is interesting to put the robust regret values into the same plots?
- Why is it claimed that the algorithm works in the continuous context regime, when the quantities defining the acquisition functions assume finiteness of the context set?
- Could you comment on the computational complexity with respect to the dimensionality or size of the input space? In the preliminaries, it is assume that the input space lives in a d-dimensional ambient space. Does d play any role in quantities estimated in the paper?
- The paper would benefit from an elaboration on the significance of the regret bound. In this context, why is the robust regret a meaningful metric to measure the success of the procedure? Considering the definition of the DRBO problem, it might be beneficial to rather consider the (robust) objective value error of the last iteration T.
- "In particular, we also present a similar analysis, showing that the robust regret decays sublinearly for the right choices of radii." It is unclear to me why the regret bound from Theorem 2 corresponds to any type of decay.

**Limitations:**

The authors mention the question of the appropriate choice of the distributional robustness distance for specific settings as future work.

---

> ### Author Rebuttal · Authors · 2023-08-08
>
> Thank you for your review and pointing out the computational superiority of our method with the phi-divergence generalization. Indeed, while our work attempts to alleviate the finiteness of contexts assumption, we provide some clarification and discussion below regarding the discretization argument which will be added to the updated version - we appreciate your concern in bringing this up.
> ___
> Question: “Do we need to know the values of , the reference distribution $p_t$, the kernel?”
>
> Answer: Yes, these are required to compute the quantities.
> ___
> Question: “Why is the maximum information gain defined differently than in Kirschner et al.? Elaborate.”
>
> Answer:  The minor difference between maximum information gain in our paper and Kirschner et al. is the noise variance term $\sigma{-2}$ for which we follow the original maximum information gain presented in Srinivas et. al.
> ___
> Question: “Why is it claimed that the algorithm works in the continuous context regime, when the quantities defining the acquisition functions assume finiteness of the context set?”
>
> Answer:  The theorem we develop makes no finiteness assumption of the context space and so in theory when using the proposed objective, we will be robust to other distributions with continuous support. As you mention we discretize to estimate the expectation and variance terms in practice, which seemingly appears to violate this. However, noting that since the acquisition functions are all bounded (by a constant M, which is standard as in BO), an elementary application of McDiarmind’s inequality [1] allows us to bound the difference between the discretized variation and continuous one by a factor of $O\left(\frac{M}{\sqrt{n_D}}\right)$ (with high probability) where $n_D$ is the number of discretizations we perform. Therefore, the discretization forms a very close approximation to the true quantities in Theorem 1 and can apply to continuous context regimes such as $C = [0,1]$. We remark that a similar argument can be applied to Kirschener et. al. regarding continuous contexts however will require solving a linear program whose variable size is of $O(n_D)$, and thus a linear program solver will be very expensive. Therefore, the number of discretizations in their work cannot be too large. In contrast, we can perform a much more fine-grained discretization since we have a simple expression for our DRO objective, which can be computed in linear time $O(n_D)$. Thank you for this point, we believe this discussion will clarify and strengthen the contribution.
>
> [1] Doob, J. L. (1940). "Regularity properties of certain families of chance variables" (PDF). Transactions of the American Mathematical Society. 47 (3): 455–486. doi:10.2307/1989964. JSTOR 1989964.
> ___
>
> Question: Could you comment on the computational complexity with respect to the dimensionality or size of the input space? In the preliminaries, it is assumed that the input space lives in a $d$-dimensional ambient space. Does $d$ play any role in quantities estimated in the paper?
>
> Answer: the optimization complexity grows with the input dimension d. When d increases, our regret bound is loose. In Theorem 2, when $d$ increases, the maximum information gain \gamma_t (defined in Line 267, 268) will increase with d and thus the upper bound will worsen.
> ___
> Question: It is unclear to me why the regret bound from Theorem 2 corresponds to any type of decay.
>
> Answer: We refer to this as "decay" since if one chooses $\varepsilon_t = \Gamma_{\varphi}^{-1}\left(\frac{1}{\sqrt{t} + \sqrt{t+1}}\right)$, then the overall regret becomes will yield a rate of $\sqrt{T}$ which is considered state-of-the-art.
>
> ___
> Question: “The paper would benefit from an elaboration on the significance of the regret bound. In this context, why is the robust regret a meaningful metric to measure the success of the procedure? Considering the definition of the DRBO problem, it might be beneficial to rather consider the (robust) objective value error of the last iteration $T$. “
>
> Answer: we have followed the literature in BO/DRO to use the cumulative robust regret as a metric. Thank you for the suggestion. However, we think the robust objective value at the last iteration T may not be appropriate because the last point is not necessarily the optimal solution of the problem and we do not pick the last point, i.e. $x_T$ as the final $\arg\max f(\cdot)$.
> ___
>
> We feel we have addressed all of the concerns raised. If this is not the case, please let us know so that we can have the opportunity to discuss further.

---

> > ### Comment · Reviewer_NSv3 · 2023-08-16
> >
> > Thank you for your reply, which clarifies some of the questions I had satisfactorily.
> >
> > I maintain my rating as after reading the authors' rebuttal, my main concern about the paper's claims, the question whether the considered DRO-BO problem is equivalent to a finite-dimensional optimization problem even in the continuous context setting, has not been positively clarified.
> >
> > Furthermore, my question
> > > The motivation of the comparisons of the different DRBOs in Figure 2 and Figure 3 is currently unclear: The different curves correspond to different settings (balls of distributions with respect to different distance notions). Why is interesting to put the robust regret values into the same plots?
> > remained unaddressed.

---

> > > ### Author Response · Authors · 2023-08-17
> > >
> > > Dear Reviewer NSv3,
> > >
> > > Thank you for letting us know that we have clarified some of the questions satisfactorily. We below address two remaining questions.
> > >
> > > ---
> > >
> > > Regarding our claim of "infinite dimensional optimization problem reducing to a finite dimensional variable", we mean this in terms of the size of optimization variables (while Q is infinite dimensional, only $\lambda$ and $b$ remained in the reduced problem). However, in the event where $p_t$ is continuous and we discretize $p_t$ or when $p_t$ is finitely supported, the Theorem still reduces the optimization variable number from the size of the support (which can be large for many samples/discretizations) to just $\lambda$ and $b$. This is a significant improvement computationally compared to [1]. Thank you for pointing this out as this key advantage should be highlighted in the paper..
> > >
> > >
> > > ---
> > >
> > >
> > > Regarding the question: The motivation of the comparisons of the different DRBOs in Figure 2 and Figure 3 is currently unclear: The different curves correspond to different settings (balls of distributions with respect to different distance notions). Why is it interesting to put the robust regret values into the same plots? remained unaddressed.
> > >
> > >
> > > Apologies for overlooking this question in the first response. Thanks for bringing it up again and giving us the opportunity to clarify.
> > > In making the comparison in Fig 2 and 3, we follow the literature in DRBO [1] to compare the performance using the robust regret over iteration axis. Comparing different optimization algorithms or settings based on their performance over iterations helps in identifying which algorithms are more efficient for a particular problem, e.g., some algorithms might converge faster initially but slow down later, while others might converge more steadily throughout the optimization process.
> > >
> > >
> > > Note that this way of comparison is very popular in the Bayesian optimization community [2] – the primary setting considered in our paper. Having said that, we are also open to your suggestion on the alternative comparison. What would be a better choice for comparison across the DRBO methods?
> > >
> > > ---
> > >
> > > We thank you again for your time, we hope we have been able to convince you of our contributions. Otherwise please let us know, if there is any concern left.
> > >
> > >
> > > [1] Kirschner, Johannes, et al. "Distributionally robust Bayesian optimization." International Conference on Artificial Intelligence and Statistics. PMLR, 2020.
> > >
> > > [2] Srinivas, Niranjan, et al. "Gaussian process optimization in the bandit setting: no regret and experimental design." Proceedings of the 27th International Conference on International Conference on Machine Learning. 2010.

---

### Official Review · Reviewer_DEDT · 2023-07-06

**Soundness:** 3 good
**Presentation:** 3 good
**Contribution:** 3 good
**Rating:** 8
**Confidence:** 4

**Summary:**

In this paper the authors extend the domain of distributionally robust bayesian optimization (DRBO) as introduced by Kirschner et al. to the case of distributions with continuous support. The focus on the case of $\phi$-divergences and show that for these problem then DRBO problem can be reformulated in closed form using the convex conjugate of the function $phi$. They then focus on the case of $\chi^2$-divergence and the total variation metrics and show that the distributionally robust reformulations for these problems are equivalent to regularization problems. For the general case they also provide a bound on the robust regret, The conclude by illustrating their methods through numerical experiments.



**Strengths:**

The results provided by the authors are novel extend the applicability of distributionally robust optimization significantly and as such form a significant contribution.
The paper is well written and clearly illustrates the key concepts.

**Additional Comments**

Most existing work that looks at distributionally robust optimization with continuous support focuses on problems with finite distributions. In this paper the authors have focused on the case of continuous support which is significantly more challenging. For this setting they have provided a general robust reformulation along with specific reformulations for specific uncertainty sets. These new reformulations cane be optimised by straightforward first order methods without compromising on the structure of the original problem. They is a key novel contribution of their work.



**Weaknesses:**

The numerical results while interesting didn't seem to address the benefits of extending the results to continuous support as compared to discrete support. Since this is a key contribution of this paper it would be better to see results illustrating the benefit of this.

**Additional Comments**

As I said in my previous review, the numerical experimented presented by the authors seem to present a general comparison against robust methods. While this is okay, I feel it misses the key point of the work which I feel to be the reformulation of DRO problem with continuous support. As such I would have been better for the paper if the authors had numerically illustrated the benefit of such a reformulation over a continuous set. For example, is reformulation over this continuous support set better then simply reformulating over a finite set created using samples.



**Questions:**

1. I would be interested in knowing if the authors have tried to identify the structure of the worst case distribution.
2. It would also be interesting to know how many of the $phi$-divergences can be converted into regularization problems.

**Limitations:**

Not applicable.

---

> ### Author Rebuttal · Authors · 2023-08-08
>
> Thank you for your positive review and noting our key contributions. Regarding our numerical experiments: since existing methods only focus on finite context sets, we showcase the benefits of our methods on such datasets. We then present an additional experiment on a continuous context where $p_t$ is selected to be uniform over $[0,1]$ where we do show that indeed, reformulating over continuous support sets is better than simply creating a finite set using samples.
>
> ---
>
> Question: I would be interested in knowing if the authors have tried to identify the structure of the worst case distribution.
>
> Answer: In the case of KL-divergence, the worst-case distribution will resemble an exponential family whose base measure is center distribution which has been studied in the context of label-shift such as in [1]. For general $\varphi$-divergences, we suspect the distribution will be some generalization of this under differentiable assumptions on $\varphi$.
>
> ---
>
> Question: It would also be interesting to know how many of the $\phi$-divergences can be converted into regularization problems.
>
> Answer: This is a very good question since under the assumption of $\varphi$ being twice differentiable, a well-known study on $\varphi$-divergences (see [2] and Remark 4 of [3]) has shown that $\varphi$ (via Taylor expansion) can be approximated with the chi-squared divergence. Since the chi-squared divergences yields a variance regularization term, an approximation argument under the assumption that $\varphi$ is twice-differentiable, will lead to regularization. Therefore, we conjecture under smooth choices of $\varphi$, DRO-BO admits a regularization interpretation. Thank you for this point, we will add this discussion into the paper.
>
> ---
>
> [1] Zhang, J., Menon, A., Veit, A., Bhojanapalli, S., Kumar, S., & Sra, S. (2020). Coping with label shift via distributionally robust optimisation. arXiv preprint arXiv:2010.12230.
>
> [2] Nielsen, F., & Nock, R. (2013). On the chi square and higher-order chi distances for approximating f-divergences. IEEE Signal Processing Letters, 21(1), 10-13.
>
> [3] https://people.lids.mit.edu/yp/homepage/data/LN_fdiv_short.pdf

---

> > ### Comment · Reviewer_DEDT · 2023-08-11
> > **Response.**
> >
> > I thank the authors for their response. I will maintain my review as it is.

---

### Official Review · Reviewer_xPHX · 2023-07-09

**Soundness:** 3 good
**Presentation:** 3 good
**Contribution:** 2 fair
**Rating:** 5
**Confidence:** 2

**Summary:**

This work studies distributionally robust Bayesian optimization (DRO-BO) problems with $\varphi$-divergences which cover $\chi^2$-divergence, total variation distances and KL divergence. The authors show that the minimax DRO-BO problem has an equivalent minimization problem, and propose an algorithm for solving the special cases of $\varphi$-divergences. They complement their theoretical results with numercial experiments comparing against existing methods.

**Strengths:**

* This work presents an interesting perspective of DRO-BO by offering a reformulation into an equivalent minimization problem through convex conjugates, which opens up possibilies of efficient algorithms for solving DRO-BO.

**Weaknesses:**

I have no major concerns.

**Questions:**

N/A.

**Limitations:**

Yes, the authors have addressed the limitations according to the checklist.

---

> ### Author Rebuttal · Authors · 2023-08-08
>
> Thank you for your time in reviewing and positive comments towards our work.

---

### Official Review · Reviewer_Yhe4 · 2023-07-09

**Soundness:** 4 excellent
**Presentation:** 3 good
**Contribution:** 3 good
**Rating:** 7
**Confidence:** 3

**Summary:**

The paper proposes a new approach for Distributionally Robust Bayesian Optimization. The paper address the problem of data shift in phi divergence which generalizes better than previously studied types and subsumes other known divergences categories including chi^2 divergence, Total Variation, and the extant Kullback-Leibler. The paper proposes new expressions for acquisition functions under two types of divergences ( chi^2 divergence, Total Variation), A theoretical analysis showing the problem reduction to a minimization problem, and a regret-bound analysis.

**Strengths:**

+ The paper provides a theoretical analysis that reduces the computationally intractable problem of data shift in the context of BO to a tractable simple optimization problem.

+ The paper is overall well-written and self-sufficient. The preliminaries section covers the needed technical definitions for the rest of the paper for readers unfamiliar with the problem details. Writing can be enhanced by adding comparison and discussion about contextual BO.

+ Interesting and technically solid paper. The final acquisition function expressions are simple yet the impact and generality of the approaches to the new types of divergences are important. I consider this an advantage since it becomes more amenable to execution.

+ The paper provides an adaptive expression for the acquisition function hyperparameter \epsilon leading to a hyperparameter "free" approach when there is no prior knowledge from the user about the suitable value of \epsilon.

**Weaknesses:**

+ There is common motivation with contextual BO beyond the robustness literature. This is clear in the motivation but seems to be ignored later as relevant methods for comparison both on the qualitative and quantitative levels. Please address this part. I believe it is crucial and might be confusing to readers who are familiar with only one line of work and not the other.

+ The experimental setup is limited in terms of baselines and applications:
      +  The baselines from Regular BO included only one acquisition function.
      +  The StableOpt baseline was omitted from the power wind experiment and computational time comparison.
      +  There are only two synthetic experiments and two real-world experiments.


**Questions:**

+ It is not clear if, for the experiments in figures 3 and 4, the \epsilon was varied based on its theoretical expression or was set to a fixed value in advance

+ Is there a reason for omitting some of the baselines including StableOpt from the wind power experiment and the computational time comparison?

+ Why is the impact of the hyperparameter C studied only for the wind power experiment?


Please refer to the weaknesses

**Limitations:**

Limitations are discussed

---

> ### Author Rebuttal · Authors · 2023-08-08
>
> Thank you for your review and inclination to accept this paper. We will include additional motivation for contextual BO and hope that the below answers address your concerns regarding the experiments.
>
> ---
>
> Question: “It is not clear if, for the experiments in figures 3 and 4, the \epsilon was varied based on its theoretical expression or was set to a fixed value in advance”
>
> Answer: Indeed, the \epsilon is set following the theoretical expression (defined in Lines 273-274) in Figures 3 and 4.
>
> ---
>
> Question: “Is there a reason for omitting some of the baselines including StableOpt from the wind power experiment and the computational time comparison?”
>
> Answer: We omit the plot for StableOpt in the Wind Power to avoid occlusion from the plots as well as focusing on the behaviors of different DRBO variants.
>
> ---
>
> We will include the computational time comparison for StableOpt which relies on the worst outcome which can be readily estimated from the GP surrogate model. Therefore, StableOpt is quite efficient, like the standard BO, in terms of computational complexity.

---

> > ### Comment · Reviewer_Yhe4 · 2023-08-12
> >
> > Thank you for answering my questions. I think StableOpt should be included in the comparison for the wind power experiment especially since it is the only real-world experiment. Based on the current figure, it does not seem that adding it will cause any occlusion.

---

> > > ### Author Response · Authors · 2023-08-14
> > > **Further response to Reviewer Yhe4**
> > >
> > > We thank the Reviewer for the suggestion of including StableOpt for the Wind Power experiment.
> > >
> > > We have run this experiment and get the figure with StableOpt ready. As per NeurIPS instruction, the authors are not allowed to add the external link to the figure during the response and the option of uploading the PDF file is expired after 9th August.
> > >
> > > In general, our DRBO approaches still perform better than StableOpt which only looks at the worst-case scenarios.
> > >
> > > In the final version of the paper, we will update the Figure 4 including StableOpt for Wind Power experiment.

---

> > > > ### Comment · Reviewer_Yhe4 · 2023-08-17
> > > >
> > > > Thank you for your response. Usually, in the case of no figure allowed, a table with intermediate results over different iterations can be added. Please note that even if the results are negative, it would be valuable to share them as they will open the discussion of when and why would each method work best.

---

> > > > > ### Author Response · Authors · 2023-08-18
> > > > > **Table with results over iterations**
> > > > >
> > > > > We thank the review for the follow up response. As suggested, due to no figured allowed, we have presented below the intermediate results over iterations for Wind Power dataset with different context size |C| with different methods.
> > > > >
> > > > >
> > > > > ## Wind Power |C|=30
> > > > >
> > > > > | Iteration | 0      | 100| 200| 300| 400| 500| 600| 700|
> > > > > |-----------|--------|--------|--------|--------|--------|--------|--------|--------|
> > > > > |DRBO $\chi^2$ | 0.02 (0.16) | -0.99 (0.31) | 0.44 (0.42) | 1.89 (0.66) | 4.49 (0.83) | 8.19 (0.86) | 11.62 (0.92) | 14.73 (0.96) |
> > > > > |DRBO TV| 0.02 (0.16) | -0.01 (0.40) | 0.11 (0.45) | 1.33 (0.81) | 3.71 (0.97) | 7.27 (1.08) | 10.59 (1.26) | 13.65 (1.32) |
> > > > > |DRBO KL| 0.02 (0.16) | -4.77 (0.63) | -5.33 (0.87) | -3.19 (0.86) | 1.09 (0.78) | 5.74 (0.75) | 9.00 (0.86) | 12.63 (0.94) |
> > > > > |DRBO MMD| 0.02 (0.16) | -2.07 (0.60) | -3.01 (1.50) | -1.55 (1.56) | 1.47 (1.60) | 5.21 (1.65) | 7.99 (1.48) | 10.89 (1.33) |
> > > > > |BO| 0.02 (0.16) | -0.85 (0.47) | -0.42 (0.70) | 1.68 (0.72) | 4.11 (0.75) | 6.66 (0.75) | 9.10 (0.75) | 11.09 (0.74) |
> > > > > |StableOpt| 0.02 (0.16) | 0.08 (0.55) | 0.81 (0.80) | 2.35 (0.98) | 4.48 (1.14) | 7.21 (1.31) | 9.24 (1.49) | 11.00 (1.58) |
> > > > >
> > > > >
> > > > >
> > > > > ## Wind Power |C|=100
> > > > >
> > > > >
> > > > > | Iteration | 0      | 100| 200| 300| 400| 500| 600| 700|
> > > > > |-----------|--------|--------|--------|--------|--------|--------|--------|--------|
> > > > > |DRBO $\chi^2$ |  0.02 (0.16) | 0.24 (0.39) | 0.88 (0.53) | 2.59 (0.78) | 5.52 (0.94) | 9.42 (0.99) | 13.18 (1.03) | 16.36 (1.07) |
> > > > > |DRBO TV|  0.02 (0.16) | 0.08 (0.45) | 0.67 (0.40) | 2.87 (0.62) | 5.79 (0.77) | 9.93 (0.80) | 13.86 (0.87) | 17.27 (0.93) |
> > > > > |DRBO KL| 0.02 (0.16) | -4.54 (0.64) | -4.68 (0.84) | -3.55 (1.02) | 0.03 (1.10) | 4.52 (1.29) | 8.66 (1.36) | 12.83 (1.37) |
> > > > > |DRBO MMD| 0.02 (0.16) | -1.13 (0.63) | -0.82 (0.72) | 1.02 (1.30) | 4.23 (1.78) | 8.08 (1.86) | 11.53 (1.90) | 14.41 (1.90) |
> > > > > | BO | 0.02 (0.16) | -0.85 (0.47) | -0.42 (0.70) | 1.68 (0.72) | 4.11 (0.75) | 6.66 (0.75) | 9.10 (0.75) | 11.09 (0.74) |
> > > > > |StableOpt| 0.02 (0.16) | 0.21 (0.55) | 1.11 (0.80) | 2.76 (0.98) | 4.98 (1.14) | 7.79 (1.31) | 9.95 (1.49) | 11.70 (1.58) |
> > > > >
> > > > > ## Wind Power |C|=500
> > > > >
> > > > >
> > > > > | Iteration | 0      | 100| 200| 300| 400| 500| 600| 700|
> > > > > |-----------|--------|--------|--------|--------|--------|--------|--------|--------|
> > > > > |DRBO $\chi^2$ |  0.02 (0.16) | -0.46 (0.28) | 0.63 (0.36) | 2.27 (0.61) | 5.03 (0.77) | 8.84 (0.79) | 12.60 (0.81) | 15.78 (0.83) |
> > > > > |DRBO TV|  0.02 (0.16) | -0.57 (0.31) | 0.99 (0.42) | 2.46 (0.58) | 5.61 (0.71) | 9.89 (0.73) | 13.85 (0.77) | 17.33 (0.80) |
> > > > > |DRBO KL|  0.02 (0.16) | -4.48 (0.70) | -5.28 (0.94) | -3.86 (0.90) | -0.09 (1.00) | 4.67 (1.06) | 8.49 (1.09) | 12.83 (1.15) |
> > > > > |DRBO MMD| 0.02 (0.16) | -0.93 (0.34) | -0.14 (0.39) | 2.57 (0.57) | 6.10 (0.55) | 9.91 (0.54) | 13.17 (0.70) | 16.24 (0.70) |
> > > > > |BO| 0.02 (0.16) | -0.85 (0.47) | -0.42 (0.70) | 1.68 (0.72) | 4.11 (0.75) | 6.66 (0.75) | 9.10 (0.75) | 11.09 (0.74) |
> > > > > |StableOpt| 0.02 (0.16) | 0.24 (0.55) | 1.20 (0.80) | 3.02 (0.98) | 5.31 (1.14) | 8.15 (1.31) | 10.28 (1.49) | 12.07 (1.58) |

---

> > > > > > ### Comment · Reviewer_Yhe4 · 2023-08-21
> > > > > >
> > > > > > Thank you for providing the results.

---

### Official Review · Reviewer_1hSk · 2023-07-10

**Soundness:** 3 good
**Presentation:** 3 good
**Contribution:** 2 fair
**Rating:** 6
**Confidence:** 3

**Summary:**

This paper provides distributional robustness in the context of the Bayesian Optimization (BO) problem. Although there is existing work in this field, such work is rudimentary, and the authors develop a more general theory that works with generic $\varphi$-divergence-based ambiguity sets. The proposed algorithm works with familiar $\varphi$ functions, and furthermore, the authors derive closed-form expressions specifically for total variation, $\chi^2$, and KL-divergence functions. They then adopt existing Gaussian process-based BO solvers for the function evaluations in the reformulated expressions and derive a sublinear regret bound which differs from the existing BO regret bounds due to a new term that is the "price of distributional robustness".

**Strengths:**

I think the paper is clear and the reader is not getting distracted. The language is clear. There are no under/overpromises. The proofs are correct to my understanding. I believe the proof of Theorem 2 is sound. I also like the discussion after Theorem 1 on how the Variance term can be related to the existing DRO papers that relate distributional robustness to some variants of regularization.

**Weaknesses:**

The largest weakness in my view is the lack of a DRO literature review. The current literature review is based on BO, but regardless of what structure $f$ has (black-box or anything else really), there are thousands of papers out there, and especially $\varphi$-divergence is an overstudied topic. I find it hard to be convinced that Theorem 1 is useful or novel. Almost none of the papers reviewed in the "Distributionally Robust Convex Optimization" paper by Wiesemann, Kuhn, and Sim are cited.

Moreover, half of the paper is on DRO, but I don't see a connection to BO. It looks like there are two separate fields, and the connection is light. Especially Algorithm 1, if I am not wrong, already standard in the BO literature and the contribution looks like the derivation of $\alpha(x)$.

I have further suggestions and questions below.

I am giving a slight acceptance decision conditional on a more thorough literature review in the rebuttal period that would potentially convince me and the readers. Except for that, I would like to thank the authors for this work and the clean paper.

**Update:** The score is updated from 5 to 6. Please see the discussions.

**Questions:**

- In the abstract and the following, could the authors please elaborate on `sublinear regret' -- sublinear in what? There are contexts and ambiguity (hyper)parameters.

- Page 2, citation 48 is missing brackets.

- Page 2, "as one would expect": why? There are many studies showing it does not necessarily give complicated problems.

- Page 2, "several baselines": could the authors please be more specific?

- Page 2: I don't think the first contribution listed is new. It is already established that the inner problem can be dualized in the majority of the literature.

- Whenever "performs best empirically" is mentioned, could the authors please specify if this is out-of-sample?

- In general, "why distributional robustness" instead of robust optimization (e.g., in the GP setting one can also think $c$ additively perturbed) is not discussed. I see the relevance, but just a discussion could be useful.

- Similarly, why $\phi$-divergences but not Wasserstein-balls? The former has some difficulties in real-life low-data settings due to the support constraints.

- Section 3, "receives a context $c_t$": could the authors please clarify that this is independent of $x$?

- Page 3, "full uncertainty information with any prediction": not clear

- Footnote 2: please define which the "majority of considered examples" are.

- Equation (3): firstly it is said that the interest of DRO is to "compute" the function, but isn't it to "optimize" it? There might be an $\inf$ missing here. I don't follow why the expectations are indexed by $q(c)$ but not $q$. Afterward, it is said to be intractable, but again, there are many traceability results under different assumptions -- this is not a thorough summary.

- Please also define that $\mathbb{E}$ is over the empirical distribution; otherwise, the problems on page 4 do not have a meaning.

- $p_t$ is the reference distribution: maybe state that it is the empirical distribution and give some consistency properties of this.

- Page 4, $B_{\varphi}^t(p)$: is $p$ supposed to be $p_t$?

- Theorem 1, "measurable": according to which measure?

- Theorem 1: can you please add a discussion of whether there is anything used about BO (I think not)? If not, then please let the user know that this formulation still needs computation of $f$ and for this purpose, you will revise the UCB-related algorithms from the literature.

- Page 5, "existing BO advancements": please cite.

- Examples 1-2: are these simply replacing the conjugate or are there further steps? Would be great to clarify.

- Example 2: Could the authors please comment on the computational complexity of this problem?

- Example 1, "very easily implemented": this reads a little subjective here. Please try to formalize.

- Page 6, "convenient in the theoretical analysis": perhaps it should be 'convenience'?

- I would recommend having more discussion on the second term of Theorem 2 and try and bring some insights.

- Minor: In the appendix, there is a part where there is "to to" twice.

- Why are the KL divergence results pushed to the appendix? I would have thought for most people that could be the most interesting divergence.

**Limitations:**

The limitations are clearly addressed in the paper.

---

> ### Author Rebuttal · Authors · 2023-08-08
>
> Thank you for your review and generally positive inclination of our paper regarding the clarity and novelty of our work. We apologize for the lack of references with respect to DRO and we will fix this. Indeed, Theorem 1 at a technical level may not be novel however its application Bayesian Optimization certainly is in this current form. We will cite all work from "Distributionally Robust Convex Optimization"  and references therein.
>
> ---
>
> Question: In the abstract and the following, could the authors please elaborate on `sublinear regret' -- sublinear in what? There are contexts and ambiguity (hyper)parameters.
>
> Answer: Here, we are referring to sublinear with respect to the number of iterations T similar to Kirschener et. al, Thank you for bringing this to our attention as our bounds have several other parameters at play.
>
> ---
>
> Question: Page 2, "as one would expect": why? There are many studies showing it does not necessarily give complicated problems.
>
> Answer: Indeed, however we only mean it is complicated if we are solving the DRO problem in BO exactly and directly by tackling the minimax problem since one needs to minimize while also maximizing the objective - this could lead to an unstable solution.
>
> ---
>
> Question: In general, "why distributional robustness" instead of robust optimization (e.g., in the GP setting one can also think  additively perturbed) is not discussed. I see the relevance, but just a discussion could be useful.-divergences but not Wasserstein-balls? The former has some difficulties in real-life low-data settings due to the support constraints.
>
> Answer: Yes this is a fair point. We focus on robustness at the distributional level and  phi-divergences largely due to their smooth properties such that we can exploit Fenchel duality.
>
> ---
>
> Question: “is $p$ supposed to be $p_t$”
>
> Answer: Yes, this is a typo.
>
> ---
>
> Question “Theorem 1, "measurable": according to which measure?”
>
> Answer: This is respect to the Borel measure so measurability becomes quite a mild condition.
>
> ---
>
> Question: “Theorem 1: can you please add a discussion of whether there is anything used about BO (I think not)? If not, then please let the user know that this formulation still needs computation of f and for this purpose, you will revise the UCB-related algorithms from the literature.
>
> Answer: Indeed, there is nothing specific to BO here; however the results specific to UCB and BO in combination with Theorem 1 appear in our regret analysis in Theorem 2. Thank you for pointing this out, we can have more of a discussion there stating this fact.
>
> ---
> Question “Examples 1-2: are these simply replacing the conjugate or are there further steps? Would be great to clarify.
>
> Answer: Yes these are specifically replacing the conjugates and further simplifying the expressions.
>
> ---
> Question: Example 2: Could the authors please comment on the computational complexity of this problem?
>
> Answer: For the acquisition function, this is equivalent to Equation (5) where one needs to find the min and max values across all observed contexts, and therefore is linear in the number of observed contexts.
>
> ---
> Question: “Why are the KL divergence results pushed to the appendix? I would have thought for most people that could be the most interesting divergence.”
>
> Answer: While our result does apply to the KL divergence, we get readily available closed forms for chi-squared and total variation as seen by the second term which forms a regularization term, which the KL divergence does not reduce to.
>
> ---
> Thank you for your additional comments on improving clarity and presentation. We agree that it would be interesting to have a discussion around the regularization term and the effect it was. We will also pay more respect to the DRO results as you have stated, and hope that we have addressed your concerns!

---

> > ### Comment · Reviewer_1hSk · 2023-08-13
> >
> > Dear Authors,
> >
> > Thank you very much for replying to my questions!
> >
> > When you say further literature review will follow, or state "we can have more of a discussion there stating this fact", do you mean by the camera-ready version?
> >
> > Could the authors relate the results more to the DRO literature (I was hoping to see some discussion during the rebuttal period)? I still find it hard to see the novelty from the DRO side. It is OK even if the novelty is specific to BO (not a general result + a solution algorithm dedicated to BO).
> >
> > Best regards.

---

> > > ### Author Response · Authors · 2023-08-14
> > >
> > > Dear Reviewer 1hSk,
> > >
> > > Thank you for your response. When it comes to the relationship between our contributions and the DRO literature, while we use different proof techniques (such as Fenchel-duality), the results are not new for DRO. You are right that the novelty is specific to BO. From the perspective of BO however, we have several contributions:
> > >
> > > (1) We derive the acquisition function in the simple form. As the reviewer Yhe4 highlighted that this is an advantage since it becomes more amenable to execution. Our approach is general to the new types of divergences that are important.
> > >
> > > (2) We provide an adaptive expression for the acquisition function hyperparameter $\varepsilon$ leading to a hyperparameter "free" approach when there is no prior knowledge from the user about the suitable value of $\varepsilon$.
> > >
> > > (3) We derive regret bounds for DRO applied to BO with $\varphi$-divergences which are novel.
> > >
> > > We will state this fact that our work has limited novelty to DRO however focuses on BO in our camera ready version as you suggested.

---

> > > > ### Comment · Reviewer_1hSk · 2023-08-14
> > > >
> > > > Thank you very much for your replies! The Fenchel duality (and related) are also common in DRO to my knowledge, see, e.g., Theorem 3.8 and the references in https://arxiv.org/pdf/2303.03900.pdf. Please correct me if I am making a mistake.
> > > >
> > > > Your point (3) is interesting, and I see the value!

---

> > > > > ### Author Response · Authors · 2023-08-15
> > > > >
> > > > > Dear Reviewer 1hSk
> > > > >
> > > > > Indeed Fenchel duality has appeared in DRO before and your understanding is correct. We will include those references in the camera ready version.
> > > > >
> > > > > Thank you again for your time and feedback in improving our paper.

---

> > > > > > ### Comment · Reviewer_1hSk · 2023-08-15
> > > > > >
> > > > > > Thank you for your kind reply. I decided to increase my score now that I can clearly see the overall picture.

---

### Author Rebuttal · Authors · 2023-08-09

We would like to thank the reviewers for their time and efforts in reviewing our work. The reviewers are majority in acceptance of the work, as they have noticed our main contribution which is to provide “a theoretical analysis that reduces the computationally intractable problem of data shift in the context of BO to a tractable simple optimization problem.” (Reviewer Yhe4), develop an “efficient algorithm for solving DRO-BO” (Reviewer xPHX) specifically “on the case of continuous support which is significantly more challenging” (Reviewer DEDT) which “extends prior work by Kirschner et al. that considered only distributional robustness with respect to the maximum mean discrepancy.” (Reviewer NSv3). Thus the reviewers have particularly commended the novelty and clarity. However, it has come to our attention that there are certain points of clarifications regarding the discretization of contexts and missing references that the reviewers have pointed out. We will add these points and hope we have addressed the concerns of the reviewers. If this is not the case, please let us know so that we can have the opportunity to discuss further.

---

> ### Comment · Reviewer_1hSk · 2023-08-13
>
> I would like to thank the author for the summary and let them know that I am carefully reading their rebuttals to all the reviewers.

---

### Decision · Program_Chairs · 2023-09-21

**Decision:**

Accept (poster)

**Comment:**

This meta review is based on the reviews, the authors rebuttal and the discussions with the reviewers, discussions with the SAC, and ultimately my own judgement on the paper. There was a consensus that the paper contributes sound and interesting contributions. I feel this work deserves to be featured at NeurIPS and will attract interest from the community. I would like to personally invite the authors to carefully revise their manuscript to take into account the remarks and suggestions made by reviewers - in particular, making the paper more friendly to DRO researchers and highlighting their contributions in DRO and BO. Congratulations!